# Isoprene emission potentials from European oak forests derived from canopy flux measurements: An assessment of uncertainties and inter-algorithm variability

Ben Langford[1], James Cash[1, 2], W. Joe F. Acton[3], Amy C. Valach[3*], C. Nicholas Hewitt[3], Silvano Fares[4], Ignacio Goded[5], Carsten Gruening[5], Emily House[1, 2, 3], Athina-Cerise Kalogridis[6**], Valérie Gros[6], Richard Schafers[1,2], Rick Thomas[7], Mark Broadmeadow[8] and Eiko Nemitz[1]

[1] Centre for Ecology & Hydrology, Edinburgh, EH26 0QB, U.K.
[2] School of Chemistry, University of Edinburgh, West Mains Road, Edinburgh, EH9 3JJ, U.K.
[3] Lancaster Environment Centre, Lancaster University, Lancaster, LA1 4YQ, U.K.
[4] Council for Agricultural Research and Economics - Research Centre for Forestry and Wood (CREA-FL), Arezzo, Italy
[5] European Commission, Joint Research Centre, Ispra, Italy
[6] Laboratoire des Sciences du Climat et de l'Environnement (LSCE-IPSL), Unite Mixte CEA-CNRS-UVSQ (Commissariat a l'Energie Atomique, Centre National de la Recherche Scientifique, Universite de Versailles Saint-Quentin-en-Yvelines), 91198 Gif-sur-Yvette, France
[7] School of Geography, Earth and Environmental Sciences, University of Birmingham, Edgbaston, Birmingham, B15 2TT

[8] Forestry Commission, Alice Holt Lodge, Farnham, Surrey, GU10 4LH, UK

* now at: British Antarctic Survey, Cambridge, UK
** now at N.C.S.R. "Demokritos", Institute of Nuclear and Radiological Sciences & Technology, Energy & Safety, 15341 Agia Paraskevi, Attiki, Greece

*Correspondence to:* Ben Langford (benngf@ceh.ac.uk)

## Abstract

Biogenic emission algorithms predict that oak forests account for ~70% of the total European isoprene budget. Yet the isoprene emission potentials that underpin these model estimates are calculated from a very limited number of leaf-level observations and hence are highly uncertain. Increasingly, micrometeorological techniques such as eddy covariance are used to measure whole-canopy fluxes directly, from which isoprene emission potentials can be calculated. Here, we review five observational datasets of isoprene fluxes from a range of oak forests in the UK, Italy and France. We outline procedures to correct the measured net fluxes for losses from deposition and chemical flux divergence, which were found to be on the order of 5-8% and 4-5%, respectively. The corrected observational data were used to derive isoprene emission potentials at each site in a two-step process. Firstly, six commonly used emission algorithms were inverted to back out time series of isoprene emission potential, and then an "average" isoprene emission potential was calculated for each site with an associated uncertainty. We used these data to assess how the derived emission potentials change depending upon the specific emission algorithm used and importantly, on the particular approach adopted to derive an "average" site-specific emission potential. Our results show that isoprene emission potentials can vary by up to a factor of four depending on the specific algorithm used and whether or not it is used in a "big-leaf" or "canopy environment model" format. When using the same algorithm, the calculated "average" isoprene emission potential was found to vary by as much as 34% depending on how the average was derived. Using a consistent approach with version 2.1 of the Model for Emissions of Gases and Aerosols from Nature (MEGAN), we derive new ecosystem-scale isoprene emission potentials for the five measurement sites, Alice Holt, UK (10,500±2,500 µg m$^{-2}$ h$^{-1}$), Bosco Fontana, Italy (1,610±420 µg m$^{-2}$ h$^{-1}$), Castelporziano, Italy (121±15 µg m$^{-2}$ h$^{-1}$), Ispra, Italy (7,590±1070 µg m$^{-2}$ h$^{-1}$) and the Observatoire de Haute Provence, France (7,990±1010 µg m$^{-2}$ h$^{-1}$). Ecosystem-scale isoprene emission potentials were then extrapolated to the leaf-level and compared to previous leaf-level measurements for *Quercus robur* and *Quercus pubescens*, two species thought to account for 50% of the total European isoprene budget. The literature values agreed closely

with emission potentials calculated using the G93 algorithm, which were 85±75 µg g$^{-1}$ h$^{-1}$ and 78±25 µg g$^{-1}$ h$^{-1}$ for *Q. robur* and *Q. pubescens* respectively. By contrast, emission potentials calculated using the G06 algorithm, the same algorithm used in a previous study to derive the European budget, were significantly lower, which we attribute to the influence of past light and temperature conditions. Adopting these new G06 specific emission potentials for *Q. robur* (55±24 µg g$^{-1}$ h$^{-1}$) and *Q. pubescens* (47±16 µg g$^{-1}$ h$^{-1}$) reduced the projected European budget by ~17%. Our findings demonstrate that calculated isoprene emission potentials vary considerably depending upon the specific approach used in their calculation. Therefore, it is our recommendation that the community now adopt a standardised approach to the way in which micrometeorological flux measurements are corrected and used to derive isoprene, and other biogenic VOC, emission potentials.

## 1.    Introduction

Over the past 30 years much attention has been focused on understanding the processes that control emission rates of the $C_5H_8$ molecule, isoprene, from vegetation (Tingey et al., 1981; Sharkey and Loreto 1993; Guenther et al., 1993; 1995; 2006; 2012; Monson et al 1994; Goldstein et al., 1998; Petron et al., 2001; Sharkey, 2008). Isoprene is a key species in both atmospheric chemistry and climate, acting as a precursor in the formation of ground-level ozone pollution through its interactions with oxides of nitrogen ($NO_x$) and the hydroxyl radical (OH) and playing an important, but as yet, not fully quantified, role in the formation of secondary organic aerosol (SOA) (Hallquist et al., 2009; Kiendler-Scharr et al., 2009; Carlton et al., 2009). Although our understanding of why plants emit isoprene is still incomplete (Laothawornkitkul et al., 2009), robust relationships between isoprene emissions and the  photosynthetic photon flux density (PPFD) and ambient temperature have been identified and form the basis of some of the most widely used algorithms used to predict its emissions from the biosphere (Guenther et al. 1991; 1993; 2006; 2012). Although the algorithms of Guenther are perhaps the most widely used and highly cited, numerous other models exist which are formulated on a partial understanding of the underlying metabolic processes that determine production rates of isoprene synthase such as photosynthesis (Arneth et al., 2007; Niinemets et al., 1999; Martin et al., 2000; Zimmer et al., 2000; Bäck et al., 2005 and Pacifico et al., 2011).

In the Guenther algorithms, isoprene emission rates are modelled by assessing the emission potential (also referred to in the literature as an emission factor or the basal emission rate) of plant species for a set of standard environmental conditions (typically 1000 µmol m$^{-2}$ s$^{-1}$ PPFD and 303 K) which is then scaled using parameterisations of the emission response to fluctuations in light and temperature. On this basis, global biogenic isoprene emissions are thought to be on the order of 500 Tg/year (Guenther et al., 2012), accounting for around half of all non-methane VOC emissions to the atmosphere. These estimates are of course only as certain as the underpinning model parameters. Currently, the largest source of uncertainty in global isoprene emission estimates is attributed to emission potentials (Guenther et al., 2012; Arneth et al., 2008). Historically, emission potentials have been derived using enclosure measurements, where the emission rate of isoprene was measured from a single leaf or branch at standard conditions. Numerous laboratory and field studies have contributed to an extensive database of isoprene emission potentials from individual plant species which have been used to assign emission potentials to differing plant functional types (PFTs).

Keenan et al. (2009) compiled a database of leaf-level isoprene emission potentials for 80 European plant species which they used in conjunction with three separate BVOC emission models (Niinemets et al., 1999; Martin et al., 2000 and Guenther et al., 1993, 2006) to generate a comprehensive regional isoprene emission inventory for European forests. Their work highlighted the importance of oak trees, which, when averaged over the three models were shown to account for 70% of the total isoprene emissions within Europe, with the bulk (~66% of the total) attributed to just three oak species, *Quercus robur*, *Quercus pubescencs* and *Quercus petrea*. Yet, the emission potentials used in the models for these three species are based on a very limited number of leaf-level measurements and in the case of *Q. petrea*, which accounts for 16% of the total European emissions, the emission potential was taken from just a single leaf-level study. Clearly, the sparse nature of emission potential

measurements and high variability between genotypes and also leaves of the same tree (Genard-Zielinski et al., 2015) means the uncertainties associated with the isoprene emission inventory are very large (Arneth et al., 2008).

More recently, micrometeorological methods such as relaxed eddy accumulation (REA) (e.g. Olofsson et al., 2005) and eddy covariance (EC) (Karl et al., 2004; Rinne et al., 2007; Davison et al., 2009; Ruuskanen et al., 2011; Potosnak et al., 2013; Park et al., 2013; Kalogridis et al., 2014; Acton et al., 2016; Rantala et al., 2016) have been used to determine canopy-scale emissions directly. This "top-down" approach is, in principle, favourable because the flux measurements are integrated over a wide source area (the flux footprint) giving an emission potential that is representative of an ecosystem as a whole. This avoids the need to classify and measure individual emission rates for all of the species present and the effect of canopy architecture on the in-canopy profiles of temperature and radiation. In addition, micrometeorological methods do not disturb the ecosystem, avoiding the potential biases to which enclosure methods are vulnerable, and the measured emission rates are those actually leaving the canopy, net of any in-canopy losses from chemical degradation or deposition to surfaces.

While micrometeorological methods offer certain advantages over enclosure techniques they do not provide a direct measurement of the emission potential required in the emission models. Indeed, the derived standardised emission potentials are very much dependent on both the way in which the data are processed (cf. Langford et al., 2015) and the methods used to convert a measured flux into an emission potential that reflects a set of standard conditions. For example, when modelling isoprene emissions using emission potentials derived from canopy-scale measurements, large uncertainties may arise between the algorithms used in the model and for the calculation of the emission potential due to differing assumptions of the algorithms. In particular, where standard conditions are very different from the site conditions encountered during the field measurements, the model algorithms need to extrapolate over a wide range from the measurement conditions to the standard conditions for the derivation of the emission potential, and back again to the field conditions where the emissions are to be predicted, potentially using a different algorithm. This maximises the introduction of errors.

The scalability of canopy emission potentials also needs to be considered, as measurements at a given site are not necessarily transferable to similar ecosystems as the leaf area index (LAI) and canopy structure may vary significantly between locations introducing additional uncertainties (; Grote, 2007; Niinemets et al., 2010; Keenan et al., 2011).

In this study, we review (partly, previously unpublished) canopy-scale isoprene flux measurements from five European oak forests located in the UK, Italy and France. At each site we calculate the emission potential of the (sometimes mixed) ecosystem as a whole as well as the oak species separately and then interpolate our findings to the leaf-level for comparison with previous, species specific emission potentials, calculated from leaf cuvette measurements. We do this using several implementations of the most commonly used Guenther emission algorithms (Guenther et al., 1993; 2006; 2012), critically reviewing the differences observed between algorithms and the implications this might have for the modelling community. We carefully evaluate different ways in which emission potentials can be derived from micrometeorological flux measurements and quantify associated uncertainties, with the aim of guiding the community towards establishing a consistent methodology.

## 2       Method

In total, five datasets covering a total of 134 days of isoprene flux measurements made by (virtual disjunct) eddy covariance were analysed concurrently to (i) determine best practices for the processing of these data (ii) to establish robust emission potentials suitable for use in atmospheric chemistry and transport models and (iii) to compare the canopy-scale emission potentials with literature leaf-level emission potentials. These data sets comprise measurements above European oak forests in the U.K, France and Italy. All emission rates are displayed in units of μg of isoprene $m^{-2}$ $h^{-1}$ which is consistent with those used within the MEGAN model.

## 2.1    Measurement sites and datasets

### 2.1.1    Alice Holt, U.K. (AH)

Alice Holt forest is located in the south-east of England (51.17° N; 0.85° W), lying at an altitude of 80 m above sea level. The forest is dominated by oak trees (*Q. robur* with a scattering of *Q. petraea)* which are interspersed with European ash (*Fraxinus excelsior*, ~10%) a non-isoprene emitting species. The average canopy height is 20.5 m with a single sided leaf area index (LAI, $m^2 / m^2$) of 4.8. The understory comprises woody shrubs and herbs with hazel (*Corylus avelanna*) and hawthorn (*Crataegus monogyna*) being the most abundant (Wilkinson et al., 2012). Isoprene fluxes were measured between June 15 and August 16, 2005. Measurements were made from a 25 m tall lattice tower. An ultrasonic anemometer (model Solent R2, Gill Instruments) was mounted to a mast at 28.5 m and isoprene concentrations were measured using a high sensitivity proton transfer reaction – mass spectrometer (Ionicon Analytik GmbH). In total, 29 days of isoprene flux data were collected at this site. For specific details of the measurement setup the reader is referred to the Supplementary Information.

### 2.1.2    Bosco Fontana, Italy (BF)

The Bosco Fontana Nature Reserve (45.20° N, 010.74° E), is a primary old-growth semi-natural lowland oak-hornbeam forest located in the heart of the Po Valley, Northern Italy.  Pedunculate oak (*Quercus robur*), Northern Red oak (*Quercus rubra*), Turkey oak (Quercus cerris) (upper storey) and Hornbeam (*Carpinus betulus*) (under storey) are the dominant species in the forest which covers an area of approximately 2.33 $km^2$. The forest is isolated in a region now dominated by intensive agricultural and industrial activities and is one of the last remaining areas of flood plain forest in the central Po Valley.  The land immediately surrounding the forest is cultivated, becoming increasingly urbanised towards the province of Mantova 5.5 km to the south east. The forest has an average canopy height of approximately 25 m and a single sided leaf area index of 5.5 $m^2 / m^2$.

Measurements were made from a 40 m tall, freestanding, rectangular (2.5 x 3 m) scaffold structure with platforms at 8, 16, 24, 32, and 40 m. The northwest edge of the tower was instrumented with sonic anemometers and aspirated thermocouples at five heights. Eddy covariance flux measurements were made from the 32m platform using a HS-50 Gill research anemometer. A gas sampling line (PFA – OD. 18 mm ID. 13 mm) was installed and purged at ~ 60 L $min^{-1}$ from which the PTR-MS subsampled at a rate of 0.3 L $min^{-1}$. Measurements were made between June 13 – July 12, 2012 and in total, 29 days of isoprene flux data were collected at this site.

A detailed description of the instrument setup, calibration procedures and sensitivities are presented by Acton et al. (2015).

### 2.1.3    Castelporziano, Italy (CP)

The Presidential Estate of Castelporziano covers an area of about 6000 ha located along the Latium coast 25 km SW from the centre of Rome, Italy. The flux tower was located in the "Castello" experimental site (41.74° N, 12.40° E), 80m a.s.l. and 7 km from the seashore of the Thyrrenian Sea. Castelporziano has to a Thermo-Mediterranean climate with prolonged warm and dry summer periods and mild to cool winters. The soil of the experimental site had a sandy texture (sand content > 60%) with low water-holding capacity.

The experimental site is characterized by a mixed Mediterranean forest dominated by Laurel (*Laurus nobilis*) in the understory and Holm Oak (*Quercus ilex*) in the overstory. There were also large individual trees of Cork oak (*Quercus suber*) and Stone pine (*Pinus pinea*). The mean height of the overstory was 25 m, while the LAI was 4.8 $m^2 / m^2$.

Flux measurements were carried out between September 13 and October 1, 2011 from a flux tower 35 m tall. A PTR-TOF-MS (model 8000, Ionicon Analytik GmbH, Innsbruck, Austria) was housed in an air conditioned container at the bottom of the experimental tower. Air was drawn through a 1/4″ PFA-Teflon inlet tube to the PTR-TOF-MS from inlets mounted on top of the tower a few cm below a 3D sonic anemometer (Young, model 8100 VRE) at a flow rate of 18 SLM. The inlet tube

inside the container and the drift tube of the PTR-TOF-MS were heated to 50 °C to avoid condensation. No significant line artefacts of the measured BVOCs have been observed during inlet tube tests of the PFA-Teflon material used for this study. To protect the inlet line and the instruments from dust and particles, a 250 nm Teflon particle filter was mounted in front of the inlet tube. In total, 14 days of isoprene flux measurements were collected at this site. More detailed information on the experimental site and flux tower set up can be found in Fares et al. (2013).

### 2.1.4 Ispra, Italy (Ispra)

The flux station Ispra is situated in a small forest of approximately 10 ha inside the premises of the Joint Research Centre in Ispra, Italy, at the northern edge of the Po Valley (45.81° N, 8.63° E, 209 m above sea level). The forest is unmanaged since the 1950s and consists of mostly deciduous trees (*Quercus robur*, *Alnus glutinosa*, *Populus alba* and *Carpinus betulus*) with a leaf area index of 4.4 $m^2$/$m^2$ as derived from hemispheric photography during the campaign. The average height of the canopy is approximately 26 m.

Eddy covariance measurements were performed on the top of a self-standing tower 37 m above ground, using a Gill HS-100 sonic anemometer for the measurement of high frequency vertical wind velocities. Sample air was drawn from the tower top to an instrument container at the forest ground at a flow rate of 25 SLM through a Teflon tube with an inner diameter of 6 mm. Isoprene concentrations were measured from a 4 SLM sub-sample with a Fast Isoprene Sensor (Hills Scientific) located inside the air-conditioned container. Measurements were made between June 11 and August 8, 2013 and in total 54 days of isoprene flux data were collected at this site. Further details on the measurement setup are given in the Supplementary Information.

### 2.1.5 Observatoire de Haute Provence, France (O3HP)

The Oak observatory (O3HP) site is located at the Observatoire de Haute Provence (43.937° N, 5.71° E) in the heart of a 70 year old deciduous oak forest in south-east France approximately 650 m above sea level. The 5 m tall forest canopy is dominated by two species, Downy oak (*Quercus pubescens*) and Montpellier maple (*Acer monspessulanum*) which account for 75% and 25% of the foliar biomass, respectively. The understory is dominated by European smoke bush (*Cotinus coggygrian Scop.*) and a multitude of herbaceous species and grasses. The average single sided leaf area index is 2.4 $m^2$ / $m^2$. Measurements were made between June 9 – 11, 2012 and in total eight days of isoprene flux data were collected at this site. A detailed description of the site and measurements are given by Kalogridis et al. (2014) and Genard-Zielinski et al. (2015).

## 2.2 Isoprene emission algorithms

In this study we use six separate implementations of the Guenther et al. (1993; 2006; 2012) algorithms to normalise the measured fluxes to standard conditions and to assess the variation in the derived emission potentials. We also focus on the use of the algorithms in both the "big leaf" and detailed "canopy environment model" formats and discuss the performance of each. Below we provide a brief description of each of the algorithms used in this study. For further information the reader should refer to the associated citations.

### 2.21 Leaf-level algorithms

Perhaps the most widely used isoprene emission algorithm used is the leaf-level model first published by Guenther et al. (1993) hereafter termed G93.

$$F_{iso} = \varepsilon \cdot \gamma \cdot D = \varepsilon D \, \gamma_L \, \gamma_T \tag{1}$$

The algorithm assumes that the emission rate of isoprene ($F_{iso}$) from individual leaves or plants can be determined by multiplying the emission potential of the vegetation ($\varepsilon$), for a set of standard conditions (303 K and 1000 µmol $m^{-2}$ $s^{-1}$), by a scaling factor, $\gamma$ and the biomass density ($D$ in $g_{dw}$ $m^{-2}$). The scaling factor accounts for fluctuations in both light ($\gamma_L$) and

temperature ($\gamma_T$) which have been demonstrated to account for the majority of short term variation in isoprene emission rates (Guenther et al., 1991; Fall and Monson, 1992).

Isoprene emission rates from vegetation typically demonstrate a linear increase with PPFD up to a saturation point which can be described by:

$$\gamma_L = \frac{\alpha C_{L1} L}{\sqrt{1+\alpha^2 L^2}}.$$ (2)

Here, $L$ is the measured PPFD (in µmol m$^{-2}$ s$^{-1}$) and $\alpha$ (= 0.0027) and $C_{L1}$ (=1.066) are empirical coefficients, which describe the initial slope of the curve and normalise the response curve at standard conditions, respectively. These were determined experimentally based on the response curves measured in four plant species (eucalyptus, sweet gum, aspen and velvet bean). The same four species were also used to determine empirical coefficients to describe the temperature response of isoprene emissions which can be expressed as:

$$\gamma_T = \frac{exp\frac{C_{T1}(T-T_S)}{RT_S T}}{1+exp\frac{C_{T2}(T-T_M)}{RT_S T}},$$ (3)

where $T$ is the leaf temperature in K (often assumed to be equivalent to ambient air temperature), $T_s$ is the standard temperature (303 K), $R$ is the universal gas constant (=8.314 J K$^{-1}$ mol$^{-1}$) and $C_{T1}$ (=95,000 J mol$^{-1}$), $C_{T2}$ (=230,000 J mol$^{-1}$) and $T_M$ (= 314 K) are empirical coefficients.

Although this algorithm is optimised for leaf-level emissions it has proved very popular within the flux community due to its relative simplicity and has been routinely used to back out canopy-scale emission potentials based on observed isoprene fluxes (Rinne et al., 2002; Olofsson et al., 2005; Davison et al., 2009; Potosnak et al., 2013; Kalogridis et al., 2014; Valach et al., 2015; Rantala et al., 2016). In each of these studies the canopy was treated as a "big leaf" and the leaf temperature considered to be equivalent to the average air temperature. When inverting Eq. (1) to work back to a canopy-scale emission potential it is typical for the foliar density term to be removed and the canopy-scale emission potential to be reported in terms of mass per unit area of ground (rather than unit mass of biomass dry weight) per unit time which is also the convention adopted by the more recent isoprene emission algorithms.

### 2.2.2 Canopy-scale algorithms

More recently, Guenther et al. (2006; 2012) introduced the Model of Emission of Gases and Aerosols from Nature (MEGAN) which estimates isoprene emission rates based predominately upon canopy-scale isoprene emission potentials. This model represents a significant progression over the previous leaf-scale emission algorithms as it encompasses our growing understanding of the key driving environmental and meteorological variables that control the emission rates of isoprene from plants, which include the influence of both current and past light ($\gamma_l$) and temperature ($\gamma_t$), soil moisture ($\gamma_{SM}$), leaf age ($\gamma_A$) as well as the influence of the steadily increasing $CO_2$ ($\gamma_C$) concentrations in the atmosphere. Although the model takes the same basic form as Eq. (1), the MEGAN model also encompasses a detailed canopy environment (CE) model. This model accounts for the attenuation of light and temperature through the plant canopy across several discrete layers. In addition, the model also accounts for the effect of changing leaf area index (LAI) and has the flexibility to calculate emission rates based on calculated leaf temperature rather than the more commonly used air temperature,

$$\gamma = C_{CE} \cdot LAI \cdot \gamma_l \cdot \gamma_t \cdot \gamma_{SM} \cdot \gamma_A \cdot \gamma_C.$$ (4)

The increased number of gamma factors used within MEGAN inevitably means that there is an ever more detailed definition of standard conditions. Table 2 lists the standard conditions, where gamma is equal to unity, for each of the algorithms used in this study. The most noticeable difference between the original leaf-level algorithms and the MEGAN model is the change in standardised PPFD from 1000 µmol m$^{-2}$ s$^{-1}$ in the leaf level algorithms to 1500 µmol m$^{-2}$ s$^{-1}$ in the canopy scale emission algorithms. The increase in standard PPFD was made to reflect MEGANs canopy-scale approach, with the larger value thought to better replicate the solar radiation received at the top of a typical plant canopy.

In this study we use MEGAN 2.0 (Guenther et al., 2006, hereafter G06) in a "big leaf" format (e.g. the canopy is treated as a single layer and air temperature is assumed equivalent to the average leaf temperature). This method is similar to the G93 approach but incorporates a more advanced understanding of the influence of previous meteorology on current isoprene emission rates (Sharkey, 1991). This approach has previously been used to back calculate emission potentials from flux measurements made above rainforests (Langford et al., 2010) oil palm plantations (Misztal et al., 2011) and regions of California and the south-east United States (Misztal et al., 2016). As our measured fluxes are already corrected for in-canopy chemical losses and isoprene deposition we do not use the in-canopy production and loss term, $\rho$, used by Guenther et al in version 2.0 of the MEGAN model.

In our analysis we also explore the use of the more recent MEGAN 2.1 model (Excel version beta 3 provided by A. Guenther), which employs a five layer canopy environment model to better replicate the changes in isoprene emissions that occur as light and temperature are attenuated within the canopy. We utilise this model in three separate configurations which we refer to in the text as MEGAN 2.1 (a), (b) and (c). Configuration (a) is the full implementation of the model, where the air temperature is converted to leaf temperature by calculating the leaf energy balance (Goudriaan and van Laar, 1994) and the effects of both previous light and temperature are included (Sharkey, 1991; Guenther et al., 1999). Implementation (b) uses measured air temperature and assumes this to be constant with height throughout the canopy, but it still accounts for the influence of both current and previous light and temperature. The final implementation (c) uses air temperature but does not account for the influence of previous light and temperature. In each of these three configurations we do not account for the effects of varying $CO_2$ concentrations, setting it to 400 ppm, nor do we consider the effects of soil moisture. In both cases this decision was motivated by a lack of the necessary observational data across all sites. Finally, a fixed, site specific leaf area index was used within the canopy environment model for each of the three MEGAN 2.1 implementations.

### 2.2.3    The parameterised canopy environment emission algorithm

As well as the complete MEGAN 2.0 model and associated canopy environment model, Guenther et al. (2006) also developed a simplified single-layer canopy-scale representation of the full multi-layer model known as the Parameterised Canopy Environment Emission Algorithm which is designed to reduce the computational expense associated with the full model. Emission fluxes are simulated on the basis of current and past (24 h) light and temperature as well as information on the angle of solar elevation. The PCEEA approach uses a modified set of algorithms that describe the canopy-scale isoprene emission response in the absence of a full canopy environment model. Specifically, the algorithms used in the PCEEA approach are based on simulations using the full MEGAN model and canopy environment (CE) model for warm, broad leafed forests. According to Guenther et al. (2006), isoprene emission rates derived using the PCEEA approach match estimates from the full model to within 5% when applied at the global scale but may deviate by >25% at specific locations. This algorithm was used by Langford et al. (2010) to simulate isoprene fluxes in Malaysian Borneo, but the PCEEA approach performed less well than the G06 algorithm and hence was not used for the calculation of the published emission potentials.

### 2.3    Deriving emission potentials from above canopy flux measurements

Micrometeorological flux measurements of isoprene above forests allow the net mass flux into the atmosphere and its response to the driving meteorological variables to be quantified directly. In order to translate these measurements into ecosystem emission potentials for use in atmospheric chemistry and transport models it is first necessary to somehow normalise the measured fluxes to the set of standard environmental conditions used by the model, i.e. the point at which $\gamma$ equals unity. One approach is to average only those flux data recorded during periods where standard conditions were met, but in reality this may only constitute a very small fraction of the measured data. More typically, the emission potential ($\varepsilon$) is calculated by normalising the measured fluxes to standard conditions by inverting one of the emission algorithms described above. This generates a time series of isoprene emission potentials, which typically shows systematic patterns, indicating that either the

parametrisations imperfectly reflect the response of the emission to the meteorological drivers or that ε is subject to additional biological (e.g. circadian) control (Hewitt et al., 2011). Nevertheless, for the measurements to add to the emission potential database a single value needs to be chosen to represent that site. Various methods have been used to derive this single value, but there is currently no consensus in the literature as to which method is most appropriate. For example, both Misztal et al. (2011, 2014; 2016) and Langford et al. (2010) chose to derive emission potentials as the average of midday emission potentials

$$IEP = \overline{\left( \frac{F_{iso\,h_1...h_n}}{\gamma_{h_1...h_n}} \right)},$$ (5)

where $F_{iso\,h_1...h_n}$ represents the individual above-canopy flux measurements obtained between specific hours of the day ($h_1...h_n$ – typically around midday) and γ is the sum of the isoprene emission rate scaling parameters. By contrast, Rantala et al. (2016) chose to determine the emission potential as the gradient in a least squares regression between $F_{iso}$ and γ. The latter approach, which we hereafter term the LSR method, has gained in popularity (Acton et al., 2016; Valach et al., 2015; Rantala et al., 2016) with some choosing to set the intercept to zero (Kalogridis et al., 2014; Acton et al., 2016) and others leaving it to be determined by the fit (Rantala et al., 2016). Yet, the application of this approach is often questionable, because the relationship between flux and γ is sometimes non-linear and thus violates the assumptions of the least squares approach.

As we will show in this paper, although the 'average' emission potential is derived from the measurement data, over the day, the emission predicted with such 'average' emission potential does not necessarily reproduce the measured emission, because (i) the emission parameterisations are highly non-linear and (ii) the emission values observed during a day are not normally distributed. The inability of this approach to yield emission potentials that replicate the magnitude of the observed flux is a concern, especially when models are to be used for accounting purposes.

Here, we will evaluate both the average and LSR method alongside two new approaches. The first calculates the emission potential using an orthogonal distance regression (also known as a total least square regression) between $F_{iso}$ and γ. Put simply, the ODR method is a least squares regression that can be weighted based on the errors in both the dependent and independent variables. The random error of individual flux measurements determines the weighting for the fluxes, whereas constant uncertainties of ±25% and ±12% are applied to the values of γ calculated by the G93 and MEGAN emission algorithms respectively and are based on sensitivity studies by Guenther et al., (1993) and Situ et al., (2014). This, and the standard least squares regression approaches are in stark contrast to the average method which weights all data points evenly. The second approach is to use a weighted average to ensure the derived emission potential will always yield fluxes with the same average as the observed fluxes. This is calculated as

$$IEP_{weighted} = \frac{\overline{F_{iso}}}{\overline{\gamma}}.$$ (6)

This is similar to the average approach but takes the ratio of the average flux and the average of the $\gamma_i$ values rather than the average of the ratios which effectively ensures that the contribution of each single $IEP_i$ is weighted by the magnitude of the associated $\gamma_i$.

As part of this study we compare the isoprene emission potentials derived through the inversion of the most commonly used isoprene emission algorithms described above and the use of the average, LSR, ODR and weighted average methods in determining single, site specific emission potentials.

The impact of extrapolating from field to standard and back to field conditions can be minimised, by selecting a set of standard deviations that is closer to field conditions. Thus, a further strategy for the reporting of emission potentials could be to report emission potentials together with a set of reference conditions for which the emission potential is representative and then leave it to the emission modellers to either adapt their algorithm to these reference conditions or to extrapolate to their standard conditions. As this would use the same algorithm that is used for the emission calculations the errors induced by the extrapolation would cancel. This approach is explored in Section 3.4 below.

## 2.4 Accounting for dry deposition

Measurements of the emission potential made using leaf-cuvette systems on a single leaf or branch gives a direct measurement of the isoprene emission rate that inherently excludes the deposition process. By contrast, micrometeorological flux measurements reflect the net surface exchange of a compound which is a balance between the upward (emissions) and downward (deposition) mass fluxes. At our five measurement sites the flux of isoprene is dominated by the emission process so the net flux is nearly always upwards (positive), but it may still be offset slightly as some of the isoprene may undergo dry deposition to leaf surfaces. In order to calculate an emission potential that accurately reflects what is emitted directly from the vegetation it is therefore necessary to first correct measured fluxes for the effects of deposition. The dry deposition for isoprene is typically assumed to be very small and is often not corrected for, but the effects of deposition may become much more significant for other species such as monoterpenes and methanol which have been seen to be efficiently deposited to vegetation (Bamberger et al., 2011; Ruuskanen et al., 2011; Wohlfahrt et al., 2015).

Our measurements provide the average net isoprene flux for a measurement point above the tree canopy, $z_m$. The flux at the canopy surface can be defined as

$$F_s = F_m + F_d, \tag{7}$$

where $F_m$ is the measured isoprene flux and $F_d$ is the fraction that is depositing. This depositing fraction can be calculated as

$$F_d = -\frac{x_{(z_0')}}{R_c}, \tag{8}$$

where $x_{(z_0')}$ is the average concentration of isoprene at the notional (micrometeorological) average height of the exchange with the canopy, and $R_c$ is the canopy resistance. Although we did not directly measure the concentration of isoprene at the canopy top we can extrapolate our above-canopy measurements, $x_{(z_m)}$, to the surface using Eq. (9).

$$x_{(z_0')} = x_{(z_m)} + F_m(R_a + R_b) \tag{9}$$

Here, $R_a$, is the aerodynamic resistance, $R_b$, is the laminar boundary layer resistance which describes the transport through the laminar region that forms very close to the vegetation surface and both terms are calculated using direct measurements of micrometeorological parameters following Nemitz et al (2009). In the calculation of $R_b$ a value of 9.3 $\times 10^{-5}$ m$^{-2}$ s$^{-1}$ was used as the molecular diffusivity of isoprene, which was calculated using the molecular structure online calculator (EPA, 2007). Accounting for Eq. (9) the calculation of the deposition flux becomes

$$F_d = \frac{x_{(z_m)}}{R_c} + F_m\left(\frac{R_a(zm)+R_b}{R_c}\right). \tag{10}$$

The canopy resistance ($R_c$) for isoprene was set to 250 s m$^{-1}$ as experimentally determined by Karl et al. (2004) using direct measurements of isoprene fluxes above a tropical forest. This value is perhaps not ideal for use with temperate broad leaf forests and may also vary with canopy morphology and meteorological conditions. However, no further estimates of $R_c$ for isoprene could be found, highlighting the need of further research in this area.

Adding the estimate of the isoprene deposition flux to the observed net isoprene flux gives a closer approximation of what was actually released from the forest canopy but is still not the total isoprene flux as the effects of flux divergence, e.g. the chemical degradation of isoprene before it reaches $z_m$, must also be estimated and corrected for.

## 2.5 Accounting for chemical flux divergence

Flux divergence occurs when the scalar of interest is not chemically conserved during the average time it takes for transport between emission and detection at $z_m$. The magnitude of the effect is proportional to the reactivity of the compound, concentration of atmospheric oxidants (e.g. OH, $O_3$ and $NO_3$) and inversely proportional to the turbulent velocity scale which determines the rate of transport through the turbulent boundary layer, as well as measurement height. Schallhart et al. (2016) and Kalogridis et al. (2014) estimated directly the chemical loss of isoprene between canopy and measurement height to be 4% and 5% at the Bosco Fontana and O3HP sites, respectively. For the remaining sites we assume a 5% chemical loss of

isoprene which is also consistent with model simulations by Stroud et al. (2005), who predict canopy escape efficiencies for isoprene to be typically greater than 0.9.

## 2.6    Extrapolating emission potentials to different scales

The ecosystem-scale emission potentials ($\varepsilon_{eco}$) derived from the measurements were extrapolated (i) to derive the emission potential for the oak species ($\varepsilon_{can}$), correcting for the presence of other tree species, and (ii) to provide an emission potential equivalent to a leaf-level measurement ($\varepsilon_{LL}$) that could be compared to literature values. At four of the measurement sites, the only identified isoprene emitting vegetation species were oak, which meant the calculated ecosystem emission potential could be simply scaled based on the known percentage of oak present in relation to the overall tree cover. At the Ispra site the derived emission potential was a composite of the two known isoprene emitting species, *Quercus robur* and *Poplus alba* which represented 80% and 5% of the forest composition respectively. According to Keenan et al. (2009) the emission potentials of these two species on an area basis are 6,820 and 5,109 µg m$^{-2}$ h$^{-1}$ respectively. Based on the known species composition and relative emission potentials of these two species we scaled our ecosystem emission potential to assume a canopy composed of 94% oak and 6% poplar.

Leaf-level equivalent emission potentials were subsequently calculated for each site by dividing the whole-oak canopy emission potentials by values of leaf dry mass per unit area obtained from Keenan et al. (2009) for each species. This converts the canopy scale emission potentials which assume an emission rate on a per area basis to units of µg g$^{-1}$ h$^{-1}$. Leaf-level emission potentials are typically measured at a PPFD of 1000 µmol m$^{-2}$ s$^{-1}$, but in five of the algorithms we use, the standard conditions were increased to 1500 µmol m$^{-2}$ s$^{-1}$ to better replicate the solar radiation received towards the top of a tree canopy. Assessing the light response used in each model allowed us to scale $\gamma_i$ to equal one at 1000 µmol m$^{-2}$ s$^{-1}$ and ensure parity between the both the literature emission factors and those calculated using the G93 leaf-level algorithm.

## 2.7    Emission potential uncertainties

Emission potentials for VOCs are often reported without full consideration of the associated uncertainties in the derived quantity. Here, we attempt to derive an uncertainty value for all ecosystem, canopy and leaf-scale equivalent emission potentials that accurately reflects the wide range of potential uncertainties in the derived quantity.

There are several sources of uncertainty that are common across the ecosystem, canopy and leaf-scale equivalent emission potentials which include uncertainties in the normalisation of the fluxes to standard conditions, calibration gases used, the canopy resistance used in calculating losses due to deposition and the assumed in-canopy chemical loss of isoprene. Table 3 shows the known (calibration gases) and estimated (chemical loss and canopy resistance) uncertainties used at each of the five measurement sites. An isoprene gas standard was not available during the Alice Holt field measurements. Instead, concentrations were derived on the basis of the instrument transmission curve which according to Taipale et al. (2008) gives an uncertainty of approximately ±25%. The random uncertainty in derived emission potentials for each measurement site is taken as the average uncertainty of the individual flux measurements (Langford et al., 2015):

$$\overline{RE} = \sqrt{\frac{\left(\sum_{i=1}^{N} RE_i\right)^2}{N}}, \tag{11}$$

where $RE_i$ represents the individual flux measurement uncertainty and $N$ is the total number of flux measurements being averaged.

Additional uncertainties are associated with the oak specific canopy emission potentials which include the uncertainty in the species composition data and the change in LAI index that would result from assuming the canopy was comprised of only oak. Wind rose analysis of isoprene emission potentials at the Alice Holt, Ispra and Bosco Fontana sites showed variation of 14%, 19% and 28% respectively (see Supplementary Information). The comparatively short time series of isoprene fluxes at the Castelporziano and O3HP sites meant that wind rose analysis was not possible for these locations, so an uncertainty of 20%

was assigned to the species composition data. Similarly, an uncertainty of 15% was assigned to LAI data at each of the five sites. This value was then scaled based on the percentage of oak present at each site. For example, at Alice Holt the forest is 90% oak so we multiply the estimated 15% uncertainty by 1.1 to give a final uncertainty of 16.5%. By contrast, at Bosco Fontana where oak species only represent 27% of the species present an uncertainty of 26% was derived by multiplying 15% by 1.73.

In order to convert from whole-canopy to leaf-level equivalent emission potentials it is necessary to convert from an emission rate measured on a per unit area basis to an emission potential on a gram per dry leaf weight basis. The percentage leaf dry mass assumed for each oak species was taken from Keenan et al. (2009) for each of the tree species and given an assumed uncertainty of 25%. The process of converting from fluxes made on a "per area" to a "per mass" basis is clearly a source of uncertainty, but it is worth noting that this uncertainty could be eliminated if investigators making leaf-level measurements were to report their emission potentials on both a "per mass" and "per area" basis (Niinemets et al., 2011).

Finally, the total emission potential uncertainties for each site were calculated by propagating each of the uncertainties listed in Table 3 with the average random uncertainty in measured fluxes.

## 3       Results and discussion

### 3.1       Above-canopy flux measurements

The time series of the five isoprene flux data sets used in this study are shown in Fig. 1. In total 2792 hours of eddy covariance flux data were analysed and reviewed as part of the study. Isoprene fluxes were largest at the Ispra and Alice Holt forest sites with average midday fluxes of ~6,500 and 2,800 $\mu$g m$^{-2}$ h$^{-1}$, respectively. The larger emission rates reflect the canopy composition, which in both cases was > 80% oak, and the warm summer conditions. In contrast, emission rates at the Castelporziano site were comparatively small, typically below 150 $\mu$g m$^{-2}$ h$^{-1}$ despite the high temperature and high levels of solar radiation. This lower emission rate is attributable to not only a lower percentage of oak species present (27%) within the canopy, but to the particular species of oak present. At Castelporziano two evergreen oak species, *Quercus ilex* and *Quercus suber,* account for 27% of the forest canopy but both species are relatively minor emitters of isoprene (Keenan et al., 2009).

### 3.2       Comparison of averaging methods for emission potentials

Measured eddy covariance flux data from each of the five sites were normalised to standard conditions using the G93 algorithm and the MEGAN 2.1 (a) canopy-scale emission algorithm described in Section 2.2. Normalising flux data in this way effectively produces a time series of isoprene emission potentials from which a single value can be chosen that is thought to best represent the canopy. We calculated this site specific emission potential using the LSR, ODR and several variations of the average method, each described in detail in Section 2.3. For the latter approach, the time series of emission potentials were averaged over different time windows which included 08:00 to 18:00, 10:00 to 15:00, 11:00 to 13:00 and all hours.

Figures 2a and 2c show an average diurnal cycle of the isoprene emission potentials (IEPs) calculated at the Ispra forest site using the simplistic G93 "big-leaf" emission algorithm (Panel a) and the more sophisticated MEGAN model (V2.1) (Panel c). In this example, a clear diurnal pattern is visible in the isoprene emission potential calculated using the G93 algorithm. The emission potential calculated using the MEGAN model shows a slightly different evolution, with a marginal increase in magnitude from morning to evening. The amplitude of the variability in the calculated emission potential is greatly reduced compared with the performance of the leaf-level algorithm. The non-constancy of the calculated emission potentials was a feature consistent across all of our measurement sites (see Supplementary Information). There is laboratory evidence that isoprene emission potentials from some plant species are subject to circadian control (e.g. Wilkinson et al., 2006). Hewitt et al. (2011) found calculated isoprene emission potentials derived from canopy-scale flux measurements to exhibit a diurnal pattern, peaking at around midday, which they attributed to such circadian control. This assertion was later challenged by

Keenan et al. (2012) who suggested the diurnal pattern in the isoprene emission potential could be removed by tuning the light and temperature response curves of the model and its canopy model implementation to better match those typical of tropical vegetation. In either case, and regardless of its cause, a temporal trend in the emission potential means that the emission algorithm does not perfectly describe all of the factors that influence isoprene net emissions at this site.

Also shown on Fig. 2 (a and c) are the average isoprene emission potentials calculated using the LSR, ODR and average methods. For the G93 "big-leaf" algorithm (Fig. 2a) the calculated emission potentials span from ~ 5,600 to 7,900 µg m$^{-2}$ h$^{-1}$. Figure 2b shows the resulting average diurnal cycle of modelled isoprene emissions modelled using each derived average isoprene emission potential. When adopting an emission potential calculated with the widely used average approach (11:00 to 13:00) the algorithm replicates the measured average flux reasonably well at around 11 am, but it significantly overestimates

emission rates in the morning and afternoon, which is consistent with the diurnal fluctuation of the derived isoprene emission potential. The calculated emission potential decreases as the average method covers a larger proportion of the day, resulting in a significant underestimation of the measured fluxes (Fig. 2b). The isoprene emission rates simulated using the MEGAN 2.1 (a) model (Fig. 2d) are able to better replicate the observed isoprene fluxes in the morning and afternoon periods, but still overestimate fluxes when integrated across the day. The range of calculated isoprene emission potentials, 6,800 – 8,700 µg m$^{-2}$ h$^{-1}$, is smaller than that of those obtained using the G93 algorithm which reflects the reduced variability in the calculated

diurnal profile of isoprene emission potentials.

Emission potentials calculated using the LSR and ODR methods agree closely at the majority of sites, but the ODR method appears very sensitive to the magnitude of the error weighting applied. We assumed a 25% model error for the G93 algorithm, which was consistent with sensitivities studies by Guenther et al. (1993) and 12.5% for the MEGAN model (Situ et al., 2014).

For most sites these assumed model errors provide a fit and associated emission potential that is consistent with the other approaches. However, at some sites the ODR could only produce a sensible fit after adjusting the model uncertainty. For the Ispra data, for example, the MEGAN model error had to be reduced to 8% in order to produce a viable fit. The fact that manual adjustment of errors may be required with some data sets means that the ODR is unlikely to produce the consistent results required for a standardised approach.

Isoprene emission potentials were also calculated using the weighted average approach (Eq. 6). Using this method yielded emission potentials that, when used to simulate isoprene fluxes, matched exactly the integrated flux measurements. We calculated the normalised mean square error, or *M* score, between measured ($F_m$) and the modelled ($F_{mod}$) fluxes, using the different IEP methods described above to assess the performance of each.

The *M* score assess the performance of the model based on the magnitude of the overall bias, the variance of the residuals and

30 the intensity of association or correlation, with the lowest score deemed to indicate the best model performance (Guenther et al., 1993).

$$M = \frac{\overline{(F_m - F_{mod})^2}}{\overline{F_m} - \overline{F_{mod}}}$$ (12)

In the Guenther algorithms the IEP is simply a constant that is scaled in relation to the changing environmental conditions, so a change in IEP has no effect on the overall correlation between model and measurements. Therefore, in this study, relative

changes in the *M* score only reflect the magnitude of bias and bias variation. We found that the method with the lowest *M* score varied between sites and algorithms, but was most often associated with the average (11:00 to 13:00) method (see Supplementary Information). This is perhaps not surprising as fluxes were largest during midday, and thus choosing the correct IEP for those conditions resulted in the smallest M score.

The weighted average method, by definition always yielded a zero bias, but the standard deviation is typically lower than the

40 measurements (see Supplementary Information). Providing emission potentials that allow the average flux to be accurately modelled is certainly desirable, especially for regional or global VOC budget studies. Nonetheless, the use of the weighted average method might not suit all modelling scenarios. For example, local studies of atmospheric chemical process may require

simulated isoprene emissions to better replicate midday fluxes. In these limited cases the use of the average midday method might prove more suitable.

Figure 3 shows the same sets of emission potentials shown in Figure 2, but for each of the five measurement sites. Here, the emission potentials have been normalised to that derived using the weighted average method and the MEGAN model (V2.1). When plotted in this way two features become apparent. Firstly, the use of different emission algorithms to convert observed fluxes to emission potentials can result in markedly different results. This is illustrated by the divergence of open circles (G93) and closed triangles (MEGAN 2.1 (a)) and is particularly apparent at the Alice Holt and Castelporziano sites. Secondly, because the emission potential is not constant throughout the day (see Fig. 2) different averaging approaches yield very different average emission potentials even when the same algorithm is used. In these examples, the emission potential varied by as much as 30% at Alice Holt and 34% at Castelporziano. Since in the emission algorithms considered here the flux is proportional to the emission potential (Eq. 1), the same spread applies to the predicted emissions. The fact that the inferred isoprene emission potentials vary significantly by time of day is also of clear importance. Our results indicate that the derived emission potential may vary significantly depending upon the time of day the measurements were made. This is especially relevant when considering measurements made from airborne platforms or individual leaf cuvette systems that only capture a brief snap shot of the diurnal cycle. The magnitude of this effect will differ depending on the methods used, but as an example, at the Ispra site, an emission potential calculated using the G93 algorithm at 08:00 and then again at15:00 would result in values that differ by a factor of 1.5. The use of the more advanced MEGAN 2.1 model would reduce the variability marginally, but still result in emission potentials that differ by a factor of 1.45.

## 3.3 Isoprene emission potentials and inter-algorithm variability

Having established the weighted average method as the most consistent method for deriving an emission potentials that reproduces the measured average flux for a given algorithm, several isoprene emission potentials were calculated for each measurement site which reflect: (i) an actual ecosystem emission potential, (ii) an oak canopy-scale emission potential (where the emission potential is scaled to account for the known percentage of isoprene emitting species present within the flux footprint) and (iii) a leaf-level equivalent emission potential, where the whole oak canopy emission potential is converted to leaf-level based on assumed leaf biomass densities. The calculated isoprene emission potentials and their associated uncertainty are reported in tabular form in the Supplementary Information.

Ecosystem isoprene emission potentials for each of the five measurement sites are shown as the sum of the graduated bars in Fig. 4. The emission potential is divided into three parts which denote the "raw" measured ecosystem flux and the two corrections applied to this value which account for losses associated with in-canopy chemistry and the dry deposition of isoprene to the surface. The chemical loss term was ~5%, while the deposition term was calculated to be marginally larger ranging between 5 and 8% across the five sites. This value, however, remains uncertain and there is a clear need for researchers to derive accurate canopy resistance values for isoprene and other bVOCs for both temperate and tropical ecosystems.

Ecosystem emission potentials directly reflect the isoprene emitted from all of the species present within the measurement footprint. The emission is therefore not just dependent on the oak species, but also their abundance.

Consistent with this, the largest emission potentials were observed at Alice Holt which is comprised of 90% of strongly isoprene emitting oak species (*Q. robur* and a scattering of *Q. petraea*). By contrast, Castelporziano had the smallest calculated isoprene emission potentials; in addition to having only  27% oak cover, it is due to two evergreen species, *Quercus ilex* and *Quercus suber* (Fares et al., 2013), which are known to be very minor emitters of isoprene (Steinbrecher et al., 1997; Bertin et al., 1997; Owen et al., 2001). The fact that isoprene emissions can vary so dramatically within the *Quercus* genus is one of the major challenges for global BVOC emission models. Within the MEGAN framework vegetation is broken down into distinct plant functional types which are classes of vegetation that are thought to share similar biological properties and responses to

environmental drivers (Smith et al., 1997). The full MEGAN2.1 uses an isoprene emission potential map that accounts for the fraction of isoprene emitters in each landscape based on the species composition. In our single site version of MEGAN the detailed emission map is not used. Instead, 15 PFTs are used, covering land classes such as temperate and tropical forest, grasses and crops (Guenther et al., 2012). Based on the species composition data reported by Morani et al. (2014) for this site, Castelporziano maps to a blend of three PFTs: 66% "broadleaf evergreen temperate shrubs" (2,000 µg m$^{-2}$ h$^{-1}$), 6.8% "Needle leaf evergreen temperate tree" (600 µg m$^{-2}$ h$^{-1}$) and 27.3% "broadleaf evergreen temperate tree" (1,727 µg m$^{-2}$ h$^{-1}$), which represents the evergreen oak. Combining these PFTs results in an overall emission potential of 1,839 µg m$^{-2}$ h$^{-1}$ for the Castelporziano site. This value greatly exceeds the calculated emission potentials for this site, which is just 43 µg m$^{-2}$ h$^{-1}$ and serves to highlight the very large uncertainties that arise when assigning emission potentials to vegetation on the basis of plant functional characteristics.

The PFTs that describe the other four sites are also shown in Fig. 4 as a horizontal line and can be directly compared with the isoprene emission potentials calculated using the full MEGAN model (e.g. MEGAN 2.1 (a)). The sites with the highest proportion of oak provide the closest match with the PFT estimates. For example, Alice Holt, a site comprising 90% oak had an emission potential of 10,500 µg m$^{-2}$ h$^{-1}$. By contrast, the emission potential for Bosco Fontana was just 1,610 µg m$^{-2}$ h$^{-1}$ reflecting, mainly, but not fully, the much lower proportion of isoprene emitting species present (27%) at this site. To account for these differences we adjusted for the presence of non-oak tree cover to provide the emission potentials for oak only, the results are shown in Fig. 5.

The oak specific canopy emission potentials at the Ispra (9,495 µg m$^{-2}$ h$^{-1}$) and Observatoire Haute de Provence (10,654 µg m$^{-2}$ h$^{-1}$) sites now compare very closely with the broadleaf deciduous forest PFT emission potential of 10,000 µg m$^{-2}$ h$^{-1}$, and the Alice Holt and Bosco Fontana sites are also both within the range of the PFT emission potential when accounting for uncertainties. These findings suggest that the emission potentials for the "broad leaf deciduous forest" PFT are representative of canopies primarily composed of high isoprene emitting oak species such as *Quercus robur*.

At each site the derived emission potentials from the different algorithms show considerable variability, with up to a factor of 2.7 difference seen at the Bosco Fontana site. In each figure two sets of error bars are shown. The black error bars show the total uncertainty, which includes the random error as well as the systematic uncertainties from sources such as calibration gases, species composition and biomass estimates, which affect estimates at each site equally. The smaller, coloured error bars show the random error associated with the flux measurements and it is this value that should be used when comparing emission estimates at a single site for statistical differences. When viewing the emission potentials in conjunction with these errors it becomes apparent that some large statistical differences do exist between some, but not all, emission algorithms. In Figs. 4 and 5 these differences were, in part, due to the different definitions of standard conditions used between G93 and MEGAN algorithms. Yet, the leaf-level equivalent emission potentials shown in Fig. 6 have been adjusted to remain consistent with previous leaf-level observations which are typically obtained at 303 K and 1000 µmol m$^{-2}$ s$^{-1}$ PPFD. Interestingly, the G06 method (effectively the use of the MEGAN 2.0 algorithm in a "big leaf" format) yields a much lower IEP than the other algorithms at all but the Alice Holt site. This relates to the algorithms inclusion of the effects of previous light and temperature on isoprene emissions. According to Table 2, γ will equal unity only once the standard conditions are met, which in this case are a PPFD of 1500 µmol m$^{-2}$ s$^{-1}$ and 303 K for the current light and temperature and a PPFD of 200 µmol m$^{-2}$ s$^{-1}$ and 297 K for the previous 24 and 240 hours. An assessment of the previous environmental conditions at each of the five measurement sites (Figs. S5 to S9 in the Supplementary Information) reveals that the previous light and temperature regimes are typically much larger than the standard conditions. Therefore, in order to normalise the measured fluxes to standard conditions the light and temperature response curves must yield unity at much lower levels than is achieved using, for example, the G93 or MEGAN2.1 (c) algorithms, which only include responses to the current environmental conditions. Figure 7 shows the light and temperature response curves used in the G06 algorithm at each of the five sites relative to the response curves at standard conditions (black line). The largest deviations from the curves are seen in the light response (Fig. 7a), with Bosco Fontana

furthest from standard conditions, followed by Observatoire Haute de Provence, Ispra, Castelporziano and then Alice Holt. Deviations from the temperature curve are rather modest by comparison, with the largest positive deviations seen for Bosco Fontana, followed by Ispra, Castelporziano and O3HP. By contrast, data from Alice Holt are generally lower than the standard temperature response curve, which is consistent with the previous 24 and 240 hour temperature measurements at this site being typically 7 K below the standard temperature. From these curves we can conclude that the inclusion of past light and temperature conditions in the G06 "big leaf" algorithm, therefore, requires the standard response curves to increase (decrease) depending upon the relative values of the previous light and temperature and has the potential to deviate significantly to values calculated using the G93 algorithm. In our analysis the largest difference was observed at the Bosco Fontana site with the IEP calculated using the G06 algorithm over two times lower than that calculated using the G93 algorithm. From this analysis we recommend that the G06 algorithm not be applied in a big leaf format because the calculated emission potentials will likely be biased low.

Emission potentials calculated using the MEGAN 2.1 algorithms which each use a full canopy environment model were consistently larger than those calculated by the G06 "big leaf" approach. This relates to the treatment of light and temperature attenuation through the canopy which brings the previous environmental conditions in the lower layers of the canopy much closer to standard conditions. Interestingly, when the use of previous light and temperature is switched off (e.g. MEGAN 2.1 (c)) the emission potential increases as the effects of past light and temperature are no longer considered.

The fact that the different algorithms and indeed different variations of the same algorithm do not converge on a single IEP is of critical importance. It implies that VOC emission potentials reported in the literature are only representative as long as (i) they are used in conjunction with the same emission algorithm that was used to back out the isoprene emission potentials from the measured fluxes (ii) derived with an averaging method that correctly reproduces the measured flux or (iii) were measured under conditions similar to standard conditions. Using a different algorithm to simulate emission rates, or indeed a slightly different implementation of the same algorithm to that used to calculate the emission potential will clearly yield a different result. Our results show that the variations in emission potentials calculated using different implementations of MEGAN 2.1 are relatively small when changing between leaf and air temperature ($< 8.5\%$), but still marginally larger than the <5% suggested by Guenther et al. (2012), but can become much larger when the influence of previous light and temperature are ignored (45%). By contrast, differences between emission potentials calculated using the G93 algorithm and full MEGAN model can vary by more than a factor of two, even after accounting for the differing sets of standard conditions. While this level of uncertainty may be deemed tolerable for global model simulations, where other uncertainties are equally large, it may prove unacceptable for chemical transport models operating at regional or local spatial scales.

## 3.4  Reporting fluxes for defined conditions

We have demonstrated that emission potentials can vary considerably depending upon which emission algorithm is used to normalise the measured fluxes to standard conditions, especially if the standard conditions are very dissimilar from conditions encountered in the field. As already stated in Section 2.3, standard conditions are typically far removed from conditions found at many measurement sites e.g. at higher latitude sites, which means typically there is no or very little data directly measured under these conditions. One possibility to remove this uncertainty is to report an emission potential as the average flux that corresponds to a set of defined conditions encountered in the field, together with these new reference conditions.

Using a two-dimensional histogram, binning flux data by light ($\pm100\,\mu$mol m$^{-2}$ s$^{-1}$ PPFD) and temperature ($\pm\,0.5$ K) we selected the most common set of daytime conditions at each of our five measurement sites. An example histogram is shown for the Ispra forest site in Figure 8 and the average fluxes and environmental conditions that corresponded to these sets of conditions are shown in Table 4. In order to compare how emission potentials (extrapolate from the field reference conditions to the algorithm specific standard conditions) calculated using this small fraction of the available data (typically between 1.2-2%) compared with our previously calculated emission potentials, we converted the average fluxes shown in Table 4 to the

algorithm standard conditions using both the G93 and MEGAN2.1 algorithms. Using these new emission potentials, we then simulated the isoprene emission fluxes at each site and compared them to the observations.

Figure 9 shows the percentage difference between the averaged measured flux and the averaged modelled flux when using the "converted" isoprene emission potentials. Because the average predicted flux changes linearly with the emission potential, Fig. 9 implicitly also shows how these new emission factors compare with those derived with the weighted average method. The calculated bias ranged between +29% and -4% for the G93 algorithm and between +9% and -40% for the MEGAN 2.1 approaches. The bias for the G93 algorithm is typically positive which reflects the fact that the algorithm performs well at the reference conditions which represent typical daytime conditions but performs worse in the morning and afternoon, overestimating emission fluxes due to its inability to account for the attenuation of light and temperature through the canopy. The observed bias in the MEGAN2.1 simulated isoprene fluxes is driven by two factors (i) the fact that the average flux for the set of defined conditions is based on a limited number of data points (which induces a larger random error for both algorithms), ranging between $n = 4$ to $n = 19$, which may be a poor representation of the typical flux footprint and canopy heterogeneity and (ii) the defined conditions are based on current PPFD and temperature with larger uncertainty on the remaining gamma terms such as past PPFD and temperature. Therefore, we conclude that this approach succeeds in simulating emissions at 'typical' conditions encountered at each site, but less reliably reproduces the average emission.

While the reporting of fluxes at a set of defined reference conditions offers some clear advantages (e.g. the avoidance of two different algorithms being used for the forwards and backwards calculations), our analysis shows that there are also drawbacks that need to be considered. For example, here, we chose only to bin the measured flux data by the two major drivers of isoprene emissions, current light and temperature, meaning that the corresponding average isoprene emission is only suited for algorithms that use only these two variables (e.g. G93 algorithm). The more complex algorithms have many more reference parameters which means the measurement space becomes increasingly stratified, yielding far fewer flux averaging periods and resulting in larger uncertainties. In addition, with increasing bin width, additional uncertainty is introduced by averaging highly non-linear responses. We recommend future studies report both an emission potential for a set of defined conditions and an emission potential derived using the whole data set in conjunction with the weighted average method, providing a detailed description of exactly how the emission potential was calculated.

### 3.5  Comparison with literature values

The leaf-level emission potentials in Fig. 6 were compared to the literature values compiled by Keenan et al. (2009). Isoprene emission potentials derived using the G93 algorithm, which most closely replicates the standard conditions used in cuvette measurements, agree very closely with the published values. For example, *Quercus pubescens* (81 µg g$^{-1}$ h$^{-1}$) and *Quercus robur* (79 µg g$^{-1}$ h$^{-1}$) which are thought to account for some ~50% of total European isoprene emissions, had calculated emission potentials of 78±25 µg g$^{-1}$ h$^{-1}$ and 82±36 µg g$^{-1}$ h$^{-1}$, respectively, with the latter derived as the average from the Alice Holt, Bosco Fontana and Ispra forest sites. Yet, as we have stressed above, modellers must ensure that the emission potentials used in their model have been derived in a manner compatible with their emission algorithm. According to Keenan et al. (2009), the European isoprene budget was predicted using the G93 algorithm but also incorporating the effects of previous light and temperature as described by the equations in Guenther et al. (2006). This description appears consistent with the G06 approach we outline in Section 2.2.2 and we therefore also compare the published emission potentials against those derived using the G06 algorithm. Our estimates are 31% and 42% lower respectively for *Quercus robur* and *Quercus pubescens*, which, as discussed above, can be explained by the incorporation of additional standard conditions for the previous 24 and 240 hours light and temperature, which typically results in larger values for $\gamma_l$ and $\gamma_t$ and subsequently smaller emission potentials. Accounting for the lower emission potentials would see the contribution of *Quercus robur* and *Quercus pubescens* to the annual biogenic isoprene budget decrease from a combined total of 50% to 33%, which equates to an overall reduction in the

European total of ~17%. This would give a new average European isoprene budget for the period of 1960-1990 of around 0.85 Tg C a$^{-1}$.

While our analysis has focused on the calculation of emission potentials from above-canopy flux measurements and their uncertainties, it is important to recognise that the leaf-level emission potentials to which we compare are also highly uncertain. Leaf-level emission potentials vary considerably between the top and bottom of the canopy and for the same species have been shown to range between a factor of 10 (Aydin et al. 2014, van Meeningen et al. 2016) to 100 (e.g. Pokorska et al. 2011, Winer et al. 1983). Therefore, the leaf-level emission inventory compiled by Keenan et al. (2009) may not always give IEPs representative of the canopy average flux, which is directly observed by top-down micrometeorological approaches. Furthermore, leaf-level measurements are typically reported for a set of light and temperature conditions but other important environmental parameters including past light and temperature, $CO_2$ concentration and soil moisture, relevant to the more advanced emission algorithms, are typically not reported. With this in mind, we would echo the sentiments of Niinemets et al. (2011) who call for the standardisation of leaf-level measurements and would reiterate the need for both the reporting of emission potentials on both a per mass and per area basis and the inclusion of additional environmental parameters (past light and temperature and $CO_2$ concentrations) to further reduce the uncertainties introduced when comparing the performance of emission algorithms (utilising leaf level emission potentials) with above-canopy flux measurements.

## 4       Conclusions

Five sets of canopy-scale isoprene flux measurements from European oak forests have been carefully reviewed to determine new ecosystem, oak canopy and leaf-level equivalent emission potentials using different averaging techniques and six implementations of the commonly used Guenther et al. (1993, 2006, 2012) algorithms. New methods to correct derived emission potentials for the effects of chemical flux divergence and the losses of isoprene through dry deposition, two processes that are typically overlooked when determining emission potentials from micrometeorological flux measurements, have been outlined. Furthermore, we have thoroughly assessed the uncertainties in the derivation of ecosystem emission potentials and their subsequent extrapolation to whole-oak canopy and leaf-level estimates. All algorithms failed to reproduce the diurnal pattern in the flux, resulting in emission potentials being derived that apparently vary over the day, and from these various average emission potentials can be calculated, which result in mean fluxes that vary by up to a factor of two. In this study, we have chosen to calculate average emission potentials using a weighted average approach which ensures modelled fluxes share the same average as the measurements. While we believe this approach gives the most robust and reproducible assessment of the isoprene emission potential, others have used different approaches. We have shown that the isoprene emission potential can vary by more than 30% depending upon which method is used, resulting in additional, but entirely avoidable, uncertainties in emission potentials and hence modelled average emissions. We have also clearly demonstrated that for any given dataset a very wide range of emission potentials can be calculated, the values of which depend upon both the specific algorithm used and how it is implemented to back-out the emission potentials. Some of the variation between algorithms relates to changes in the standard light conditions from 1000 µmol m$^{-2}$ s$^{-1}$ PPFD in leaf-level models to 1500 µmol m$^{-2}$ s$^{-1}$ PPFD in canopy-scale algorithms. However, a comparison of the leaf-level extrapolated emission potentials which were harmonised to a similar set of standard conditions across all algorithms (e.g. 1000 µmol m$^{-2}$ s$^{-1}$ PPFD) demonstrated that these algorithms do not always yield similar emission potentials, with up to a factor of 2.7 difference. Clearly, different emission algorithms and algorithm implementations result in different emission predictions even if the same emission potentials are used, with the variability stated here. If the starting point are canopy-scale rather than leaf-level flux measurements, the emission algorithms are used twice: once for standardisation (backward calculation) and once in the model (forward calculation). If the algorithms and meteorological drivers are identical for both steps then errors in the algorithms cancel each other. By contrast, if different

algorithms are used then the uncertainties in both calculations may be additive. This is an important consideration for both the measurement and modelling community. It demonstrates the need for experimentalists to very carefully articulate exactly how published emission potentials were derived and which algorithms and in particular which parameters (e.g. past light and temperature, leaf temperature, $CO_2$, soil moisture etc.) were used to back out emission potentials. Similarly, the modelling community need to be aware of the uncertainties when using an emission potential derived using a different version, or even implementation, of the algorithm to that used in their model. Using our new, algorithm specific, isoprene emission potentials for *Quercus robur* and *Quercus pubescens* we were able to demonstrate that the previous European isoprene budget may have been systematically overestimated by as much as 17% due to inconsistencies between the emission potentials and emission algorithm used in the model. Therefore, a better estimate of the average European isoprene budget for the period of 1960-1990 is 0.85 Tg C $a^{-1}$.

In conclusion, we believe the uncertainty in isoprene emission models can be reduced by harmonising the way in which emission potentials are calculated from micrometeorological flux data. We have put forward recommendations for the extrapolation of net above-canopy fluxes back to surface emission fluxes and have outlined a new methodology to calculate the isoprene emission potential with clear justification. Nonetheless, with numerous implementations of the emission algorithms in use and their ever increasing flexibility and complexity there does not appear to be easy solution to avoid intra-algorithm biases. In the past the BVOC flux community has preferred to calculate isoprene emission potentials using the G93 emission algorithm due to its relative simplicity. Yet, our work shows that the emission potentials calculated in this way may not be compatible with more recent emission algorithms. Our recommendation is that model developers now provide single point versions of their code, as has already been done for MEGAN 2.1 (e.g. Pocket MEGAN, Excel beta 3), which can be used by experimentalists to more easily determine emission potentials from their observational data. Furthermore, we recommend that all processed canopy-scale flux data from which emission potentials are to be derived should be stored on a common community database. The VOCsNET database (http://vocsnetdata.ceh.ac.uk/) enables others to recalculate emission potentials in a fashion that is compatible for their model application and to enable re-calculation in the future to keep pace with the evolution of models such as MEGAN. All five datasets used in this study can be accessed via the VOCsNET database. In addition to the approaches of how to derive emission potentials from canopy scale flux measurements, further standardisation is also required for the micrometeorological flux measurement itself, including selection of instrumentation, instrument setup and operation, relative height of measurements above the canopy, data processing and reporting of results and uncertainties. In the near future it will also be important to ensure compatibility between traditional tower based flux observations and those made using the emerging technology of airborne eddy covariance flux measurements (Karl et al., 2009; Yuan et al., 2015; Misztal et al., 2014; Misztal et al., 2016; Vaughan et al., 2017). We believe that by developing a consistent and robust approach to calculating emission potentials from top-down flux measurements, future emission algorithms may be better parameterised through the incorporation of regional scale observations.

Whilst this analysis focused on the uncertainties involved in the reverse application of the emission algorithm to back out normalised emission potentials from canopy flux measurements, the variability between different algorithms and their implementation is the same for the forward calculation used in the emission models themselves.

**Acknowledgements**

This work was supported by the European FP7 project ECLAIRE (no. 282910) and NERC National Capability funding and inspired by discussions within the ECLAIRE community and in particular with David Simpson, Chalmers University Gothenburg. The authors thank Dr Matthew Wilkinson at Forest Research for facilitating our measurements at the Alice Holt field site. For the O3HP dataset, we thank the ANR-CANOPEE project (ANR 2010 JCJC 603 01), CNRS, CEA and the Oak observatory staff. We also thank Alex Guenther, Rüdiger Grote and two anonymous reviewers for their helpful comments and suggestions.

**Author Contributions**

E. Nemitz and B. Langford designed the research. B. Langford led the data synthesis and analysis and collected the data at the BF site. J. Cash assisted with data synthesis and performed the analysis. W. Acton and A. Valach collected and processed the BF dataset. S. Fares collected and processed the data from the CP site. I. Goded and C. Gruening collected and processed the data from the Ispra site, E. House, R. Thomas and M. Broadmeadow collected the data from the AH site. R. Schafers processed the AH data. A.C. Kalogridis and V. Gros collected and processed the O3HP dataset. B. Langford prepared the manuscript with contributions from all co-authors.

**Data Availability**

All five data sets can be accessed via the VOCsNET database hosted by the Centre for Ecology and Hydrology (http://vocsnetdata.ceh.ac.uk/)

**Competing Interests**

The authors declare no competing interests

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

**Table 1. Detailed breakdown of species composition and measurement approach for each of the five sites used in this study**

| | Alice Holt, UK | | Bosco Fontana, Italy | | Castelporziano, Italy | | Ispra, Italy | | Observatoire de Haute Provence, France | |
|---|---|---|---|---|---|---|---|---|---|---|
| **Species Composition** | *Quercus robur** (Pedunculate oak) | 90% | *Quercus robur** (Pedunculate oak) | 17% | *Laurus nobilis* (Bay tree) | 48.9% | *Quercus robur* *(Pedunculate, oak) | 80% | *Quercus pubescens** (Downy oak) | 75% |
| | *Quercus petraea** (Sessile oak) | | *Quercus cerris* (Turkey oak) | 7.1% | *Quercus ilex** (Holm oak) | 20.5% | *Alnus glutinosa* (Black alder) | 10% | *Acer monspessulanumk* (Montpellier maple) | 25% |
| | *Fraxinus* (Ash) | 10% | *Quercus rubra** (Northern Red oak) | 9.6% | *Pinus pinea* (Stone pine) | 6.8% | *Popolus alba** (White poplar) | 5% | | |
| | | | *Carpinus betulus* (Hornbeam) | 40.2% | *Quercus suber** (Cork oak) | 6.8% | *Carpinus betulus* (Hornbeam) | 3% | | |
| | | | Other | 26% | Other shrubs | 17% | Other | 2% | | |
| **LAI [m²/m²]** | 4.8 | | 5.5 | | 4.6 | | 4.4 | | 2.4 | |
| $h_c$ **[m]** | 20.5 | | 28 | | 25 | | 26 | | 5 | |
| $z_m$ **[m]** | 28.5 | | 32 | | 35 | | 37 | | 10 | |
| **Method** | vDEC – PTR-MS | | vDEC – PTR-MS | | EC – PTR-MS | | EC – Fast Isoprene Sensor | | vDEC – PTR-MS | |
| **MEGAN PFT classification** | 7 | | 7, 10 | | 1, 5, 9 | | 7 | | 7 | |

*Known isoprene emitters

LAI: single-sided leaf area index; $h_c$: canopy height; $z_m$: measurement height

**Table 2. List of standard conditions used by each of the emission algorithms in this study**

| Parameter | G93 "Big Leaf" | MEGAN2.0 "Big Leaf" /PCEEA | MEGAN2.1 + $C_{CE}$ |
|---|---|---|---|
| $\gamma_T$ [K] <br> • $T_{24}, T_{240}$ | 303 <br> - | 303 <br> 297, 297 | 303 <br> 297, 297 |
| $\gamma_L$ [µmol m$^{-2}$ s$^{-1}$] <br> • $L_{24}, L_{240}$ <br> • Sun leaves: $L_{24}, L_{240}$ <br> • Shaded leaves: $L_{24}, L_{240}$ | 1000 | 1500 <br> 200, 200 | 1500 <br><br> 200, 200 <br> 50, 50 |
| LAI [m$^2$/m$^2$] | - | - | 5 |
| $\gamma_{SM}$ [m$^3$ m$^{-3}$] | - | - | 0.3 |
| $\gamma_A$ [%] <br> • Growing <br> • Mature <br> • Old | - | - | <br> 10 <br> 80 <br> 10 |
| $\gamma_C$ [ppb] | - | - | 400 |
| $C_{CE}$ <br> • Humidity [g kg$^{-1}$] <br> • Wind speed [m s$^{-1}$] | - | - | 0.57 <br> 14 <br> 0.3 |

**Table 3. Summary of uncertainties attributed to the various steps used in the calculation of emission potentials for each of the five measurement sites**

| Site | No. data | Emission potential calculation (Eq. 11) | $R_c$* | Chemistry | Species Composition | LAI (for canopy and leaf-level emission potentials) | Leaf Dry Mass (Keenan et al. 2009) | Calibration gas (from manufacturer) |
|---|---|---|---|---|---|---|---|---|
| Alice Holt, UK | 629 | ±3% | ±25% | ±10% | ±10% | ±16.5% | ±25% | ±25%** |
| Bosco Fontana, Italy | 571 | ±25% | ±25% | ±10% | ±10% | ±26% | ±25% | ±5% |
| Castelporziano, Italy | 190 | ±16% | ±25% | ±10% | ±10% | ±26.25% | ±25% | ±5% |
| Ispra, Italy | 1226 | ±8% | ±25% | ±10% | ±10% | ±18% | ±25% | ±5% |
| O3HP, France | 176 | ±3% | ±25% | ±10% | ±10% | ±18.7%5 | ±25% | ±5% |

* $R_c$ = 250 s m$^{-1}$ (Karl et al., 2004)

** Instrument transmission efficiency used in the absence of a gas standard

**Table 4. Average isoprene emission fluxes at the Alice Holt, Bosco Fontana, Castelporziano, Ispra forest and O3HP sites under a set of defined conditions.**

| | Alice Holt | Bosco Fontana | Castelporziano | Ispra | O3HP |
|---|---|---|---|---|---|
| Average Flux [µg m$^{-2}$ h$^{-1}$] | 2143 | 1911 | 83 | 9404 | 2649 |
| σ [µg m$^{-2}$ s$^{-1}$] | 1075 | 599 | 102 | 3593 | 988 |
| $\overline{RE}$ [µg m$^{-2}$ h$^{-1}$] | 142 | 443 | 31 | 1268 | 353 |
| N [#] | 9 | 17 | 5 | 19 | 4 |
| Temperature range [K] | 293-294 | 302-303 | 300-301 | 302-303 | 294 - 294 |
| PPFD range [µmol m$^{-2}$ s$^{-1}$] | 800-1000 | 1800-2000 | 1400-1600 | 1600-1800 | 1800-2000 |
| Mean Temperature [K] | 293.4 | 302.5 | 300.5 | 302.6 | 293.7 |
| Mean PPFD [µmol m$^{-2}$ s$^{-1}$] | 915 | 1902 | 1523 | 1703 | 1852 |
| Mean 24 T [K] | 290 | 299 | 295 | 298 | 290 |
| Mean 240T[K] | 290 | 299 | 295 | 297 | 290 |
| Mean 24 PPFD [µmol m$^{-2}$ s$^{-1}$] | 432 | 680 | 424 | 556 | 625 |
| Mean 240 PPFD [µmol m$^{-2}$ s$^{-1}$] | 415 | 659 | 452 | 553 | 591 |

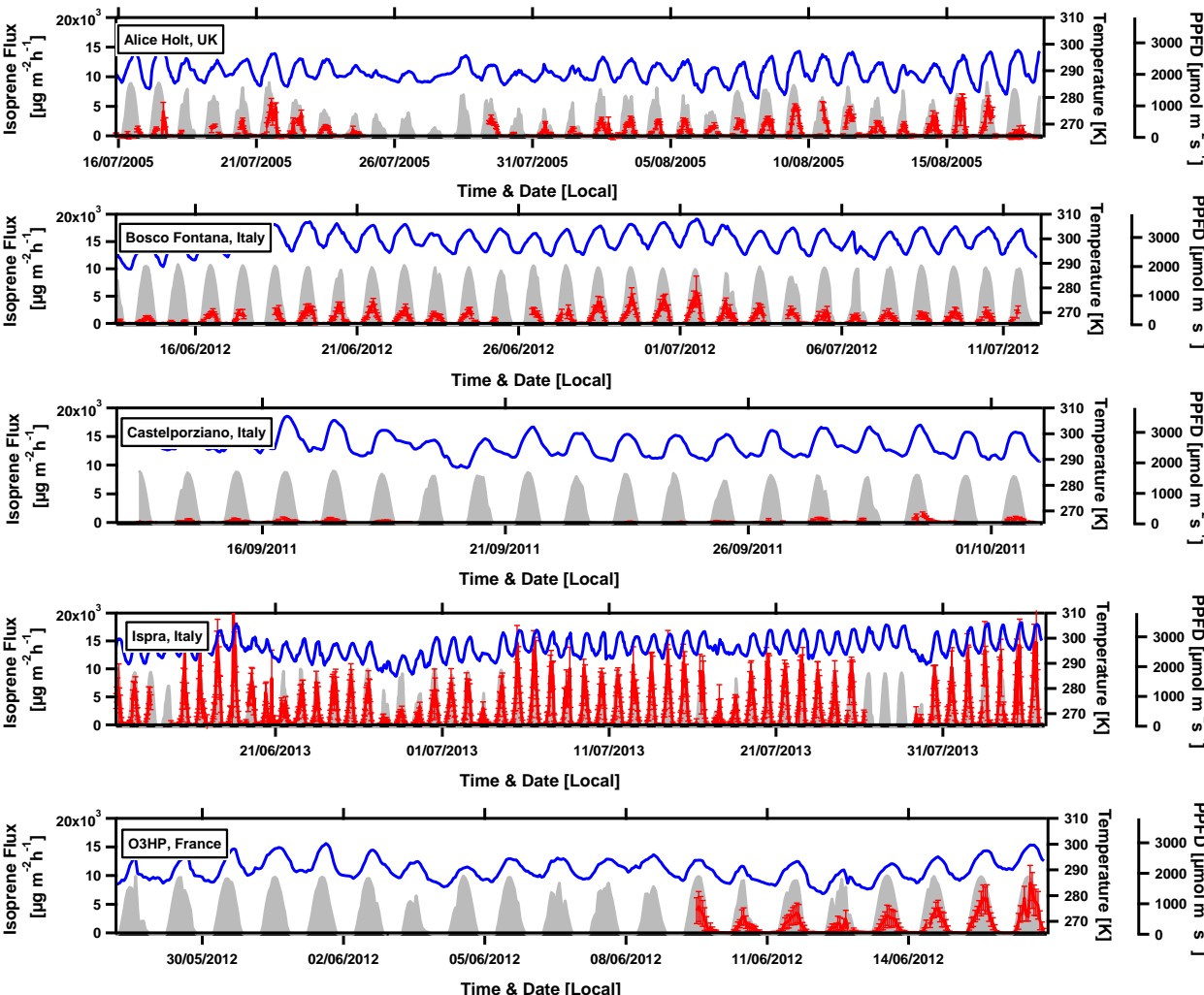

**Figure 1.** Time series of isoprene fluxes (red) in relation to temperature (blue) and PPFD (grey) at the five measurement sites. Error bars show the calculated limit of detection for each individual flux measurement.

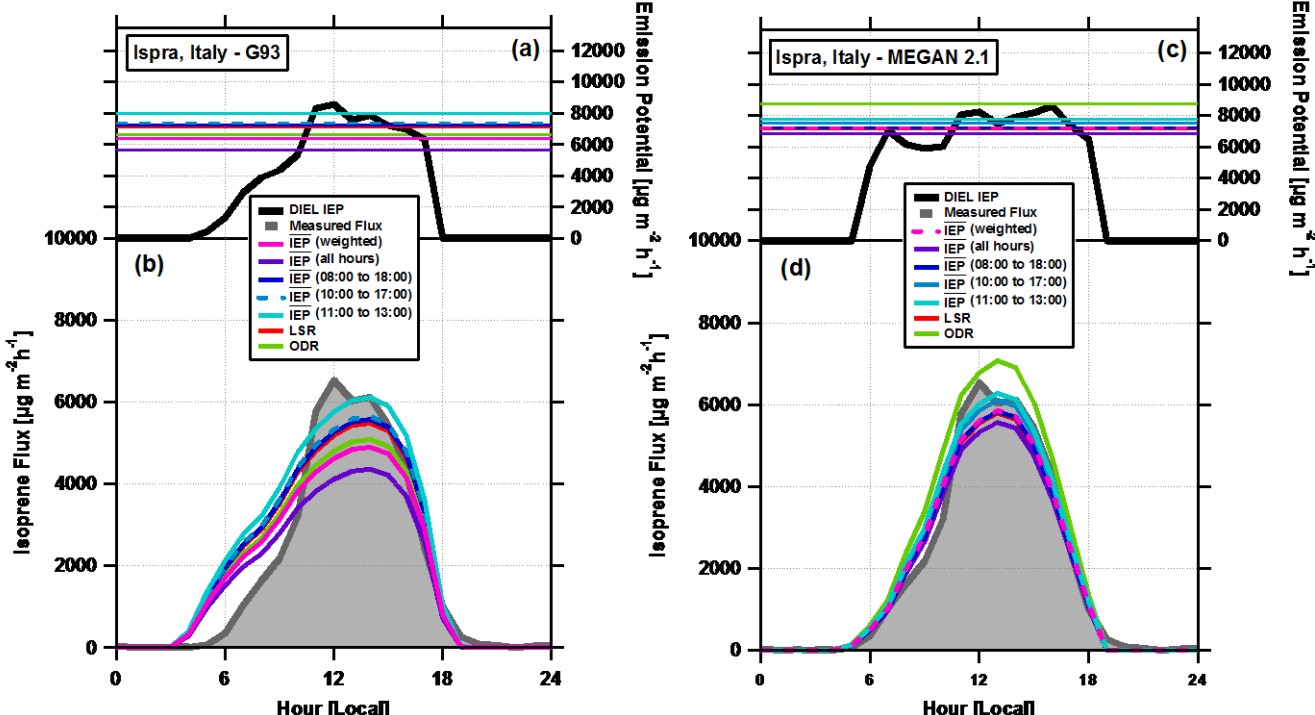

**Figure 2** Panels (a) and (c) show the average diurnal cycle in the isoprene emission potential (e.g. $IEP = \overline{\left(\frac{F_{iso}}{\gamma}\right)}$) calculated for the Ispra forest site, Italy using the G93 (panel a) and MEGAN 2.1 (panel b) algorithms. Superimposed on top of these are the isoprene emission potentials calculated using the least square regression, orthogonal distance regression and average (with several averaging lengths) methods – see text for detailed description. Panels (b) and (d) show the average diurnal cycle of the measured fluxes and the average diurnal cycle of the fluxes modelled using the seven different isoprene emission potentials calculated for this data set.

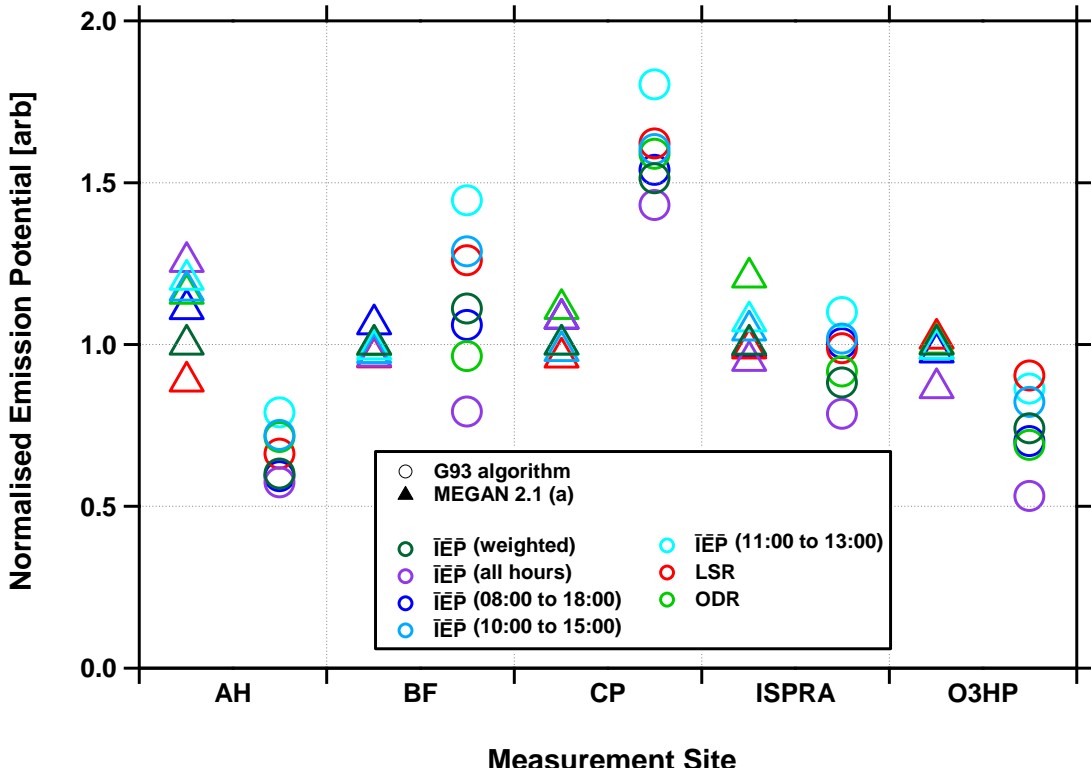

**Figure 3 Isoprene emission potentials (normalised to the MEGAN2.1 weighted average emission potential) calculated for Alice Holt (AH), Bosco Fontana (BF), Ispra forest (ISPRA) and the Observatoire de Haute Provence (O3HP). Open circles show emission potentials calculated using the G93 algorithm and closed triangles show emission potentials calculated using the MEGAN model with method (a). For each algorithm, several different approaches were used to work back to an emission potential.**

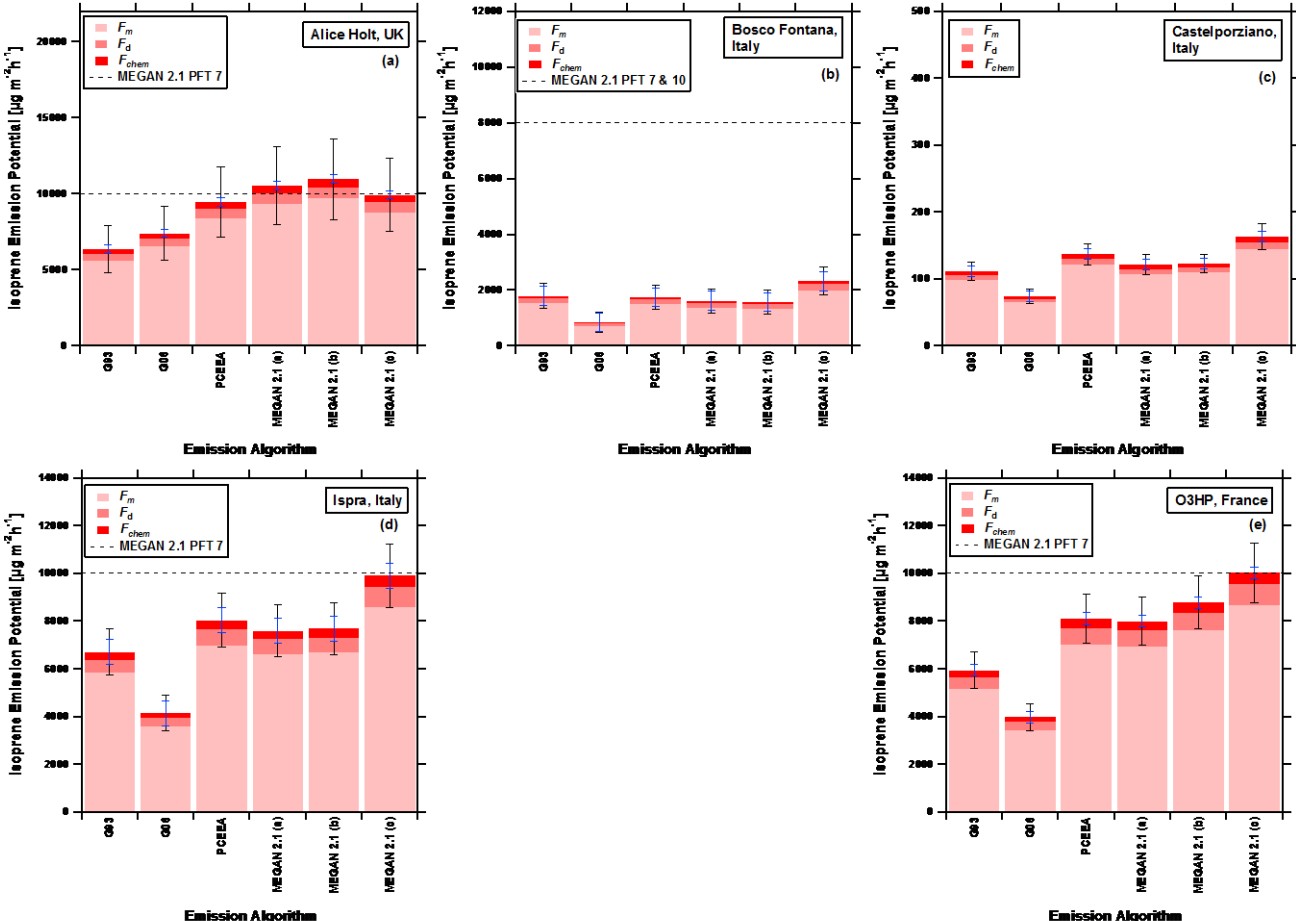

**Figure 4.** Graduated bars representing the ecosystem specific isoprene emission potentials ($\varepsilon_{eco}$) at each of the five measurement sites. Each bar shows (i) the calculated emission potential based on measured fluxes ($F_m$), (ii) the correction applied for dry deposition ($F_d$) and (iii) the correction applied for chemical flux divergence ($F_{chem}$). For each site emission potentials were calculated using six implementations of the Guenther algorithms (see Section 2.2 for details) and are shown relative to the relevant plant functional type emission potential in MEGAN 2.1 (black line). Note that for the Castelporziano site this value is at 1,839 ug m$^{-2}$ h$^{-1}$ and is off scale. The blue error bars show the uncertainty in the emission potential that relates to the random error in the observed flux measurements. The black error bars show the total uncertainty (random and systematic errors). MEGAN 2.1 (a) is the full implementation of the model using calculated leaf temperature. MEGAN 2.1 (b) is the full implementation of the model using air temperature. MEGAN 2.1 (c) is a version of the model where only the effects of current environmental conditions (e.g. light and air temperature) are used.

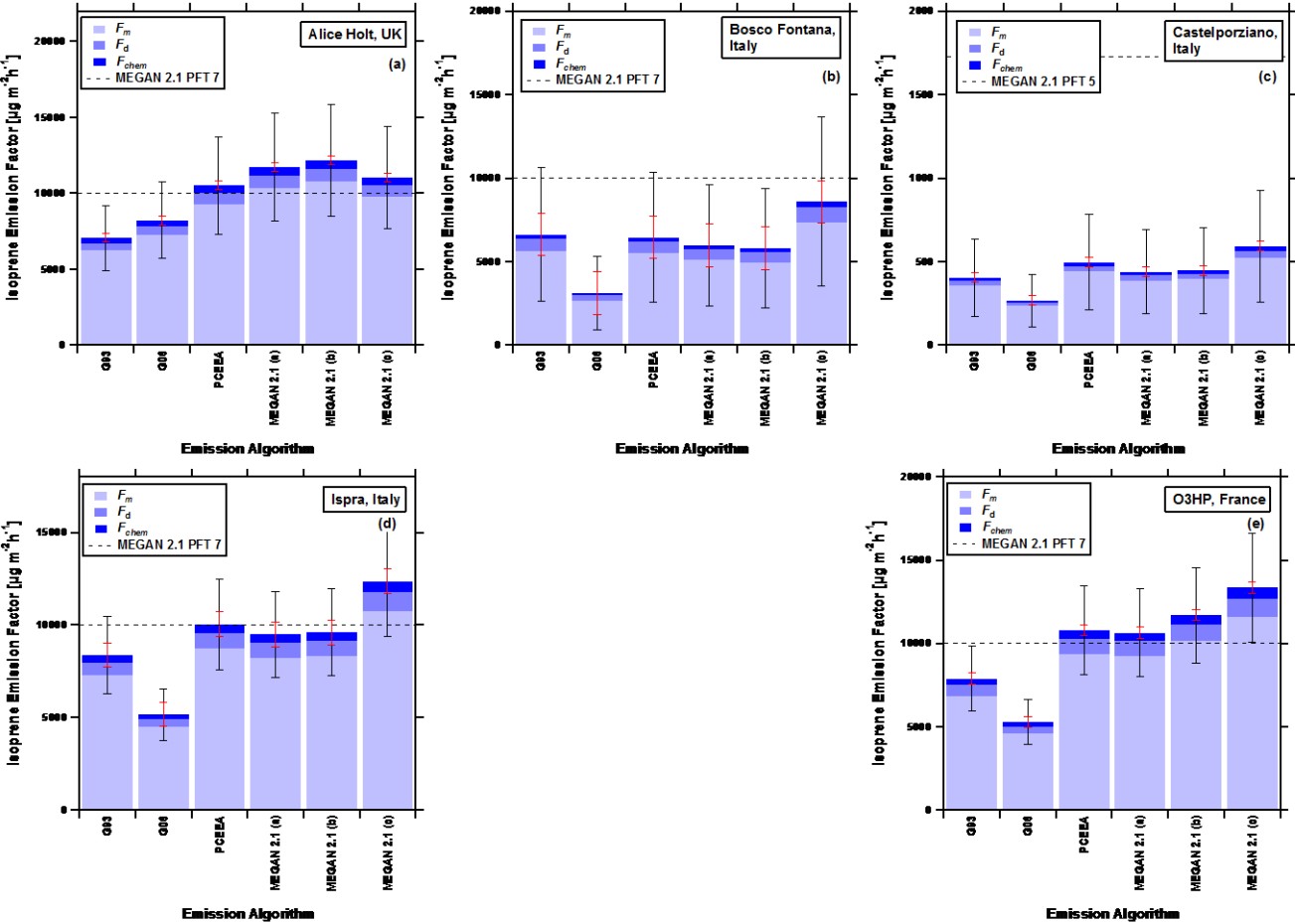

**Figure 5** Graduated bars representing an oak specific isoprene emission potentials ($\varepsilon_{can}$) at each of the five measurement sites. Each bar shows (i) the calculated emission potential based on measured fluxes ($F_m$), (ii) the correction applied for dry deposition ($F_d$) and (iii) the correction applied for chemical flux divergence ($F_{chem}$). For each site emission potentials were calculated using six implementations of the Guenther algorithms (see Section 2.2 for details) and are shown relative to the relevant plant functional type emission potential in MEGAN 2.1 (black line). The red error bars show the uncertainty in the emission potential that relates to the random error in the observed flux measurements. The black error bars show the total uncertainty (random and systematic errors). MEGAN 2.1 (a) is the full implementation of the model using calculated leaf temperature. MEGAN 2.1 (b) is the full implementation of the model using air temperature. MEGAN 2.1 (c) is a version of the model where only the effects of current environmental conditions (e.g. light and air temperature) are used.

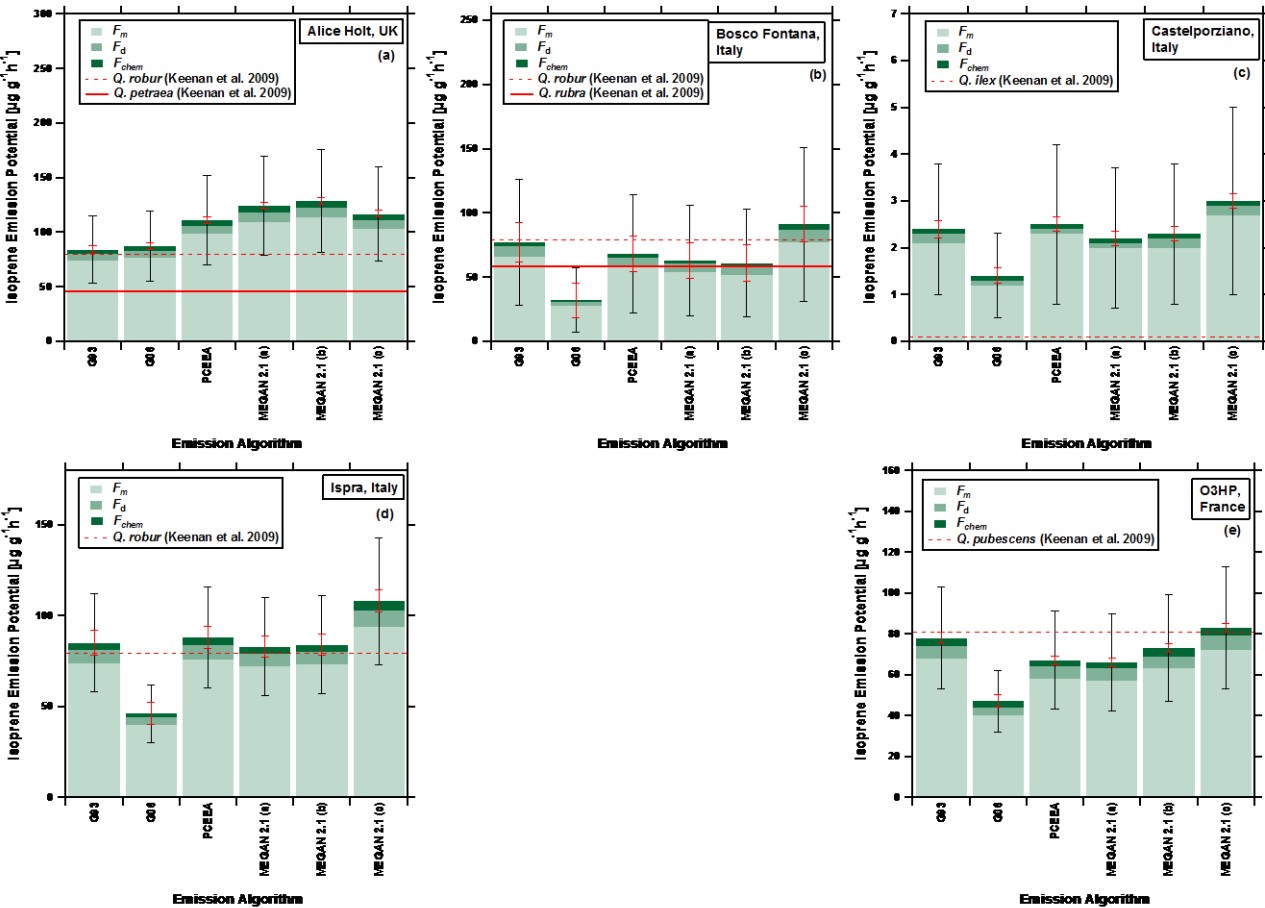

**Figure 6. Graduated bars representing leaf-level equivalent isoprene emission potentials (ε_LL) at each of the five measurement sites. Each bar shows (i) the calculated emission potential based upon measured fluxes ($F_m$), (ii) the correction applied for dry deposition ($F_d$) and (iii) the correction applied for chemical flux divergence ($F_{chem}$). For each site emission potentials were calculated using six implementations of the Guenther algorithms (see Section 2.2 for details) and are shown relative to the leaf-level emission potentials reported by Keenan et al. (2009) (red line). The red error bars show the uncertainty in the emission potential that relates to the random error in the observed flux measurements. The black error bars show the total uncertainty (random and systematic errors). MEGAN 2.1 (a) is the full implementation of the model using calculated leaf temperature. MEGAN 2.1 (b) is the full implementation of the model using air temperature. MEGAN 2.1 (c) is a version of the model where only the effects of current environmental conditions (e.g. light and air temperature) are used. All algorithms have been optimised to equal unity at 1000 µmol m⁻² s⁻¹ of PAR and 303 K.**

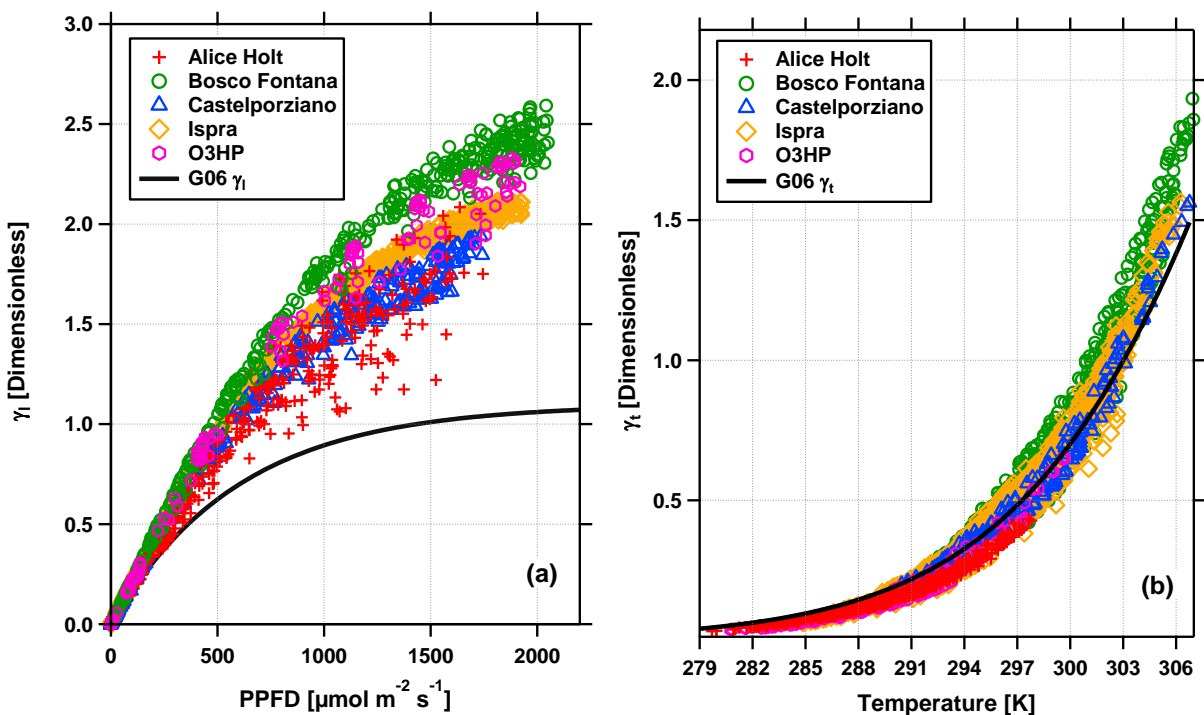

**Figure 7.** Light (a) and temperature (b) response curves from the G06 algorithm (see text for details) for the five measurement sites. The solid black lines show the light and temperature response curves when the previous light and temperature are held at standard conditions (200 µmol m$^{-2}$ s$^{-1}$ and 297 K for the previous 24 and 240 hours of light and temperature, respectively).

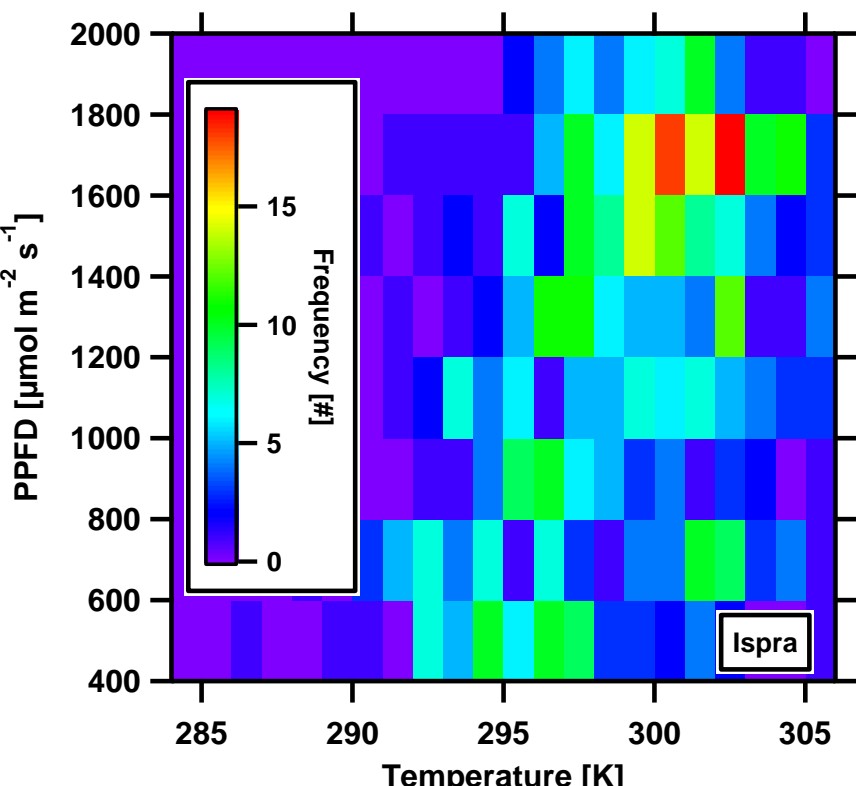

**Figure 8.** Two-dimensional histogram plot of flux averaging periods that correspond to bins of light (± 200 µmol m$^{-2}$ s$^{-1}$) and temperature (± 1 K) at the Ispra forest measurement site.

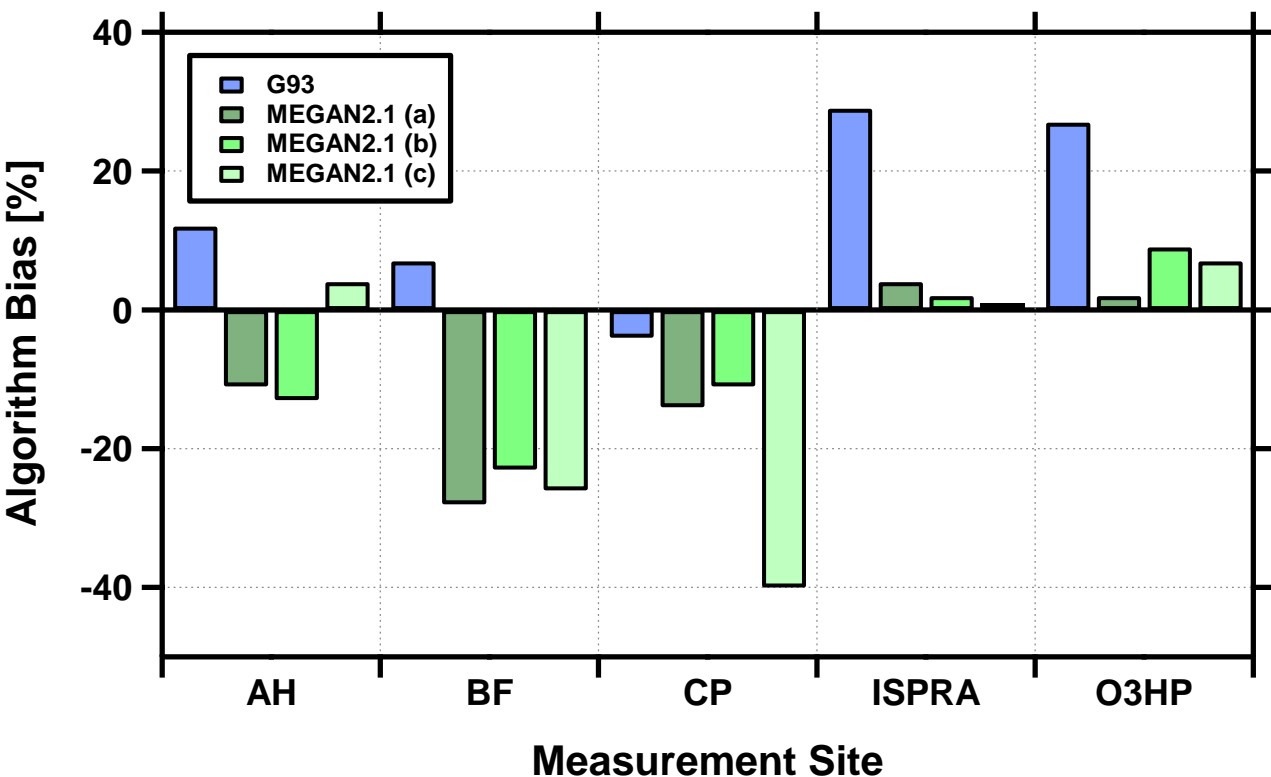

**Figure 9. Percentage bias of the average isoprene emission flux simulated by the G93 and MEGAN2.1 emission algorithms at the five measurement sites, Alice Holt (AH), Bosco Fontana (BF), Castelporziano (CP), Ispra forest (ISPRA) and the Observatoire de Haute Provence (O3HP), compared to the measured average flux.**