# Peer review of "from canopy flux measurements: An assessment of uncertainties and"

_Biogeosciences, 2017_

## Referee Comment (RC1) · A. B. Guenther (Referee) · 20 Jun 2017

This paper describes whole-canopy isoprene fluxes measured above five European forests, including both published and unpublished studies, and assesses approaches for using these measurements to parameterize the emission potential used in isoprene emission models and considers the associated uncertainties. This is an important activity that has received little attention. The paper also provides a brief mention of a community BVOC flux database that, as far as I know, has not been described in the scientific literature. The manuscript is well-written, the approach is generally valid and

the manuscript provides a substantial scientific contribution with useful guidance for several important points. I recommend publication in Biogeosciences after addressing the following comments:

General comments

1) The most important comment that I have is that the authors should consider an approach that recognizes that measurements similar to the standard conditions should be used to determine the emission potential. I recognize that this may not result in the best estimate of the daily total emission or the emission at conditions dissimilar to the standard conditions but I would argue that is not the "job" of the emission potential. The emission potential, by definition, is the emission at specified standard conditions and so the best estimate is made by either selecting measurements within some fairly narrow range of conditions or perhaps weighting measurements by how close they are to the standard conditions. It is the "job" of the emission algorithm to go from the emission potential at standard conditions to the emission at other conditions. If the emission algorithm does not do a good job of this then there will be errors but you shouldn't bias the emission potential to try account for this. Instead you should work on developing a better emission algorithm. There will be relatively little difference in the emission potential calculated by different emission algorithms if the emission potential is based on measurements made under conditions similar to the standard conditions. Of course, the problem with applying this approach to canopy scale flux data is that we can't control the measurement conditions and there may not be any that are similar to the standard conditions. If that is the case then the emission potential should be reported for some standard condition that is within the range of the observed conditions. You could then leave it up to the developers/users of a given model to convert this to an emission potential for the standard conditions of their model. Looking at Figure 1, it appears that 3 of the sites would have some measurements at T=30, PPFD=1500 while T=25, PPFD=1500 might be appropriate for the other two. You could report the emission potential for T=25 as the measured emission potential and then also report

one or more calculated emission potentials for T=30 along with an exact specification of the model approach used to get there.

2) I assume that there is at least some landscape heterogeneity at some of these sites. The authors should consider binning measurements for different "footprints" associated with different wind directions that represent different oak fractions. This could provide "replicates" with emission potentials for a larger range of oak fraction values that may provide some insights into the value and variability in the leaf-level emission potential. Of course, this assumes that there is some information on the landscape heterogeneity at these sites.

Specific comments

Page 2, line 28: "In the Guenther algorithms, isoprene emission rates are modelled by assessing the emission potential". This is not something specific to these algorithms-all isoprene emission models include some term of this type, although they may not call it an emission factor.

Page 4, line 31: delete "to" in "Castelporziano has to a Thermo-Mediterranean"

Page 6, line 18 and line 33: Be more specific about how the tendencies for studies to use a big leaf approach and using leaf temperature equals air temperature. For example, how many of the studies listed in line 16/17 do this? It may be useful to consider that at least one reason investigators do is because of the considerable effort involved in applying the full inverse canopy algorithm to their dataset and it would be useful to have an easier way to do this. For example, Yu et al. al 2017 (http://doi.org/10.1016/j.scitotenv.2017.03.262) calculated emission factors using an aircraft flux measurement dataset by using the single point version of MEGAN2.1 that you mention on page 7 line 9, and is relatively easy to use, and compare this with emission factors estimated using the regional MEGAN2.1 FORTRAN code, which is relatively difficult to use. A possible recommendation from your study is that BVOC emission modelers should provide a single point version of their code that can more

easily be used to derive emission potentials from tower and aircraft flux data.

Page 8, line 32-36: The statement that "Measurements of the emission potential made using leaf-cuvette systems on a single leaf or branch gives a direct measurement of the isoprene emission rate that inherently excludes the deposition process." seems inconsistent with "but it may still be offset slightly as some of the isoprene may undergo dry deposition to leaf surfaces". The leaf cuvette measurement excludes deposition to other leaves and to the soil but there is the possibility of uptake by the emitting leaf including by phyllosphere microbes on the leaves.

Page 10, line 27: what is the basis for the 10% uncertainty assigned to species composition and 15% to LAI? Does this consider landscape heterogeneities and the uncertainty associated with differences in the LAI and species composition within the footprint of each measurement in comparison to the average for the whole area?

Page 10, line 34: The specific leaf mass that you use to convert canopy scale emission potentials to leaf-per-mass emission potentials are arguably as uncertain and variable as isoprene emission potentials. A 25% uncertainty for specific leaf mass may be a reasonable value but you should justify this number and mention how this makes it difficult to compare canopy and leaf scale measurements. This uncertainty could be eliminated if the investigators making leaf level measurements would report emissions as both "per mass" and "per area" leaf emission potentials (i.e., they should provide the specific leaf mass for each measurement) and I suggest that this be a recommendation. If some of the leaf level data that you refer to include measured specific leaf mass (and so direct measurements of per-area leaf emission potentials) then you should make this more direct comparison that does not suffer from the large uncertainties in specific leaf mass estimates (you could do this in addition to the comparison you have already made with the per-mass leaf emission potentials.

Page 11, line 29: Define what you mean by a "wide" range. The range given here of 6750 +/- 1150 is equivalent to +/- 17% which is well within the uncertainties that you

discuss. Should that be considered a wide range?

Page 12, line 22: "regional or VOC global" should be "regional or global VOC"

Page 13, line 26: MEGAN2.1 allows users the option of using a constant value for each of the 15 PFTs but the recommended approach is to use the MEGAN2.1 isoprene emission factor map that accounts for the fraction of isoprene emitters in each landscape based on plant species composition and the species specific emission potential for each location.

Page 14, line 1: The MEGAN2.1 canopy-scale emission potential for high isoprene emitters is 24000 ug m-2 h-1. The global average temperate broadleaf deciduous tree PFT isoprene emission potential of 10000 thus represents a canopy composed of 41.6% high isoprene emitting trees which is high but not "primarily composed" as stated in the text.

Page 14, line 17-22: As is pointed out in this manuscript, canopy-average leaf-level PPFD values are considerably lower than above canopy values. Even sun leaves have a PPFD that is typically 50% or less than the above canopy PPFD since they are, on average, at an angle to the sun. The MEGAN2.1 standard condition for the past 24 and 240 hour PPFD refers to the leaf-level value and it is not appropriate (i.e., it just doesn't work) to use the above-canopy value (i.e., a big-leaf model) with this equation (G06). That the G06 past 24/240 hour algorithm should not be used with the big leaf model is an important point to make in this paper but then going on to compare the MEGAN leaf-level PPFD standard condition with the measured above-canopy PPFD in figures S5 to S9 is comparing "apples and oranges" and may be confusing to some readers. It should be made clear that this is a comparison of two different parameters (above canopy PPFD and leaf-level canopy-average PPFD) and the main point is that the above-canopy value should not be used in the past 24/240 algorithm.

Page 14, line 28: Why not just conclude/recommend that the G06 algorithm should not be used with a big leaf model?

Page 14, line 23 (and Figure 1 and Figure 7): Check on the values of PPFD shown for Castelporziano. They appear to be higher than what would be expected at the top to the atmosphere. Also, note that PAR should always be expressed in units of W/m2 while PPFD is the appropriate term when you use units of micromol/m2/s.

Page 14, line 37: This sentence is confusing.

Page 14, line 40: This may be because the Castelporziano PPFD solar radiation value is incorrect as mentioned above.

Page 15, line 4-6: or when they are measured under conditions similar to the standard conditions

Page 15, line 9: Leaf-level isoprene emission potential varies considerably between the top and bottom of the canopy and also depending on the past light and temperature environment. Are the leaf level emissions representative of the canopy average, as is the case for the canopy scale measurements, and is the past light and temperature similar? If this is not known, and it is often not reported for leaf-level studies, then this point could be included in the discussion of uncertainties for this comparison.

Page 15, line 23: As discussed above, an alternative approach is to select only measurements that are close to the standard conditions.

Page 16, line 37: This is an important point and a good opportunity for you to provide some recommendations for the standardization of flux measurements.

Figures 4-6: You were generally consistent in referring to "emission potentials" but these figures refer to "emission factors". Either can be used but be consistent.

---

## Referee Comment (RC2) · Anonymous Referee #2 · 20 Jul 2017

First, let me ask what you may call a philosophical question: should the editors allow the papers which are so innovative that they change the way we think about biogenic model algorithms? I definitely think so, because this paper not only is a great overview of biogenic models but also allows to understand how the biogenic modeling approaches work when challenged against real-world ecosystem data which can now be properly inverted to back calculate emission potentials for isoprene and maybe in the future for other compounds. While the algorithms started with leaf level measurements, the paper is another strong signal that the new generation models including processbased models should be sufficiently flexible to also be capable of accepting regional parameter data. With the rising number of these ecosystem-scale measurements, this direction is well positioned to receive further refinement, improve the estimates, and most importantly to enhance process-based understanding at canopy scale. The paper is generally well written and seems incredibly useful for the biogenic flux measurement and modeling community. I would like to recommend its publication, although I do have a few rather minor comments/questions which hopefully can be addressed before publication.

**General**

1) The authors should try to avoid the confusion between the same parameters derived in a different way/scale/conditions. Alex's point to use the conditions closest to the standard conditions seems like another sensible approach worth evaluating. However, inverting the algorithm even at conditions significantly deviating from standard conditions seems still worth the exercise but must necessarily lead to larger errors from environmental parameters measured simultaneously, and potentially may become inconsistent with original model design or intent. Assuming the measured environmental parameters (e.g. T, PAR) are accurate, the value of inferring about the emission potential at different conditions seems valuable to assess how well the algorithmic activity factor works. If it works well, then the emission potential collected under the conditions close to standard should be similar to that inverted from fluxes measured at different conditions with reverse algorithm within the same footprint. For example, Figure 2 showing stable measured emission potential during the day is unbelievably encouraging, so this approach in my opinion deserves some greater attention.

2) Model parameters which were designed for the leaf level-scale may not always be compatible for comparison or extrapolation with the same parameter obtained from inverting the equation at the ecosystem scale, even if in principle it should work. For this reason, it would be helpful to use a thoughtful system of descriptors for equivalent parameters, e.g. EFcan or EFextrap, so it is clear and distinguishable how the parameter

was obtained. This will help the issue which the authors are trying to communicate to modelers (last paragraph in the abstract) that they should be careful about how these parameters were derived before using them.

3) The abstract seems somewhat heavy for a reader. The take-home message about the differences as large as a factor of four are somewhat scary and confusing. It asks for some further insight as to what exactly causes such a large difference. If you suggest the uncertainties in the inversion of the algorithms are different for different models is it because the inversion does not work perfectly or the specific algorithm does not work well for top-down inference about the emission potential (so would likely not be accurate the other way round – bottom up)? I suggest to focus in the abstract on the major points and progress, and less so much on what you did and technical detail.

**Specific**

4) P5 L33 G93 "Perhaps the most widely used" – did you mean the most highly cited?

5) P8 L4-5 Why did you leave out Langford et al. 2010 here? Misztal et al. (e.g. 2011, table 3) used approaches to estimate BER from the regression with measurements, as well as from the middle of the day (11:00 LT; which you show also here agrees well). I think you should add Langford et al., 2010 reference here, because they reported BERs as mid-day average. I would also suggest to be more neutral and refrain from subjective statements about which approaches are more popular.

6) Abstract. Seems long and overloaded. In particular the last two sentences are rather pessimistic and might agitate modelers unnecessarily, because it is hard to believe you could really be off by a factor of four if everything is done perfectly or at least it is not sufficiently clear why exactly this is the case.

7) In the concluding remarks, you focus on the way the emission potentials are derived. Do you also want to make a bigger point about how the future models could be enhanced to better assimilate observational data at regional scale?

8) It is great that you include the original definition of emission factor (collected under the standard conditions and leaf scale). I wonder if it would be worth making a distinction between the parameterized algorithm on the full-canopy observations and whether it should be labeled as the same or a modified algorithm.

9) Table 1 – since PTR-TOF-MS was used in Castelporziano, why did you write vDEC? Did you artificially convert the PTRTOF dataset to disjunct to be consistent with other measurements? Either seems fine, as long as it is clear.

10) SI S1.1 Alice Holt – Measurement setup Lag time - as the signal to noise ratio for isoprene was rather very high, why did you use the approach for low signal to noise species? Why did you not use the accurate lag-time from each half hour period?

11) SI S1.1 "to ensure the reduced electric field strength" seems somewhat random and out of context. Also 2.01 mbar suggests that the pressure was stable to 0.01 mbar. This is rarely the case. I suggest you say 2.0 mbar or 2.01+/-0.XX mbar

12) SI S.1.1 P.1 L21-22 Instead of the justification it might be appropriate just to write what the consequences are (if any). I do not think it is necessarily bad to use high resolution measurement if it is appropriately post-averaged unless it leads to counting zeros. Otherwise, can you inform what the difference is between fluxes measured at 50 ms and averaged to 200 ms, as opposed to measured at 200 ms?

13) P8 L16-30. Unfortunately, I am extremely confused by the lack of clarity here. In particular, the weighted IEP is concerning. Why do you average the activity factor across the footprints and conditions before taking the ratio? It does not seem appropriate, because, as you say, these processes are nonlinear. For example, you have to use the model to average PAR accurately. It is more intuitive to average the emission potential, because in principle it should be relatively constant for the same vegetation type (as you show in Fig. 2), and you would not have to average nonlinear processes.

14) Sect. 2.4. Isoprene deposition. Given the large gradient it is interesting that the

authors suggest the deposition can be significant even for isoprene. It would be helpful to provide the percentage range of isoprene deposition relative to total flux, in addition to canopy resistance. As Alex wisely points out, you need to be aware of epiphytic microbes whose role is not yet well understood in affecting emission and uptake of isoprene.

15) Sect. 2.6 how do you differentiate between the effective LAI and the tree cover area fraction?

16) P10 L28 As you did not calibrate isoprene on gas standard at Alice Holt, you had to estimate the concentration from relative transmission. I am generally fine with the approach, but it should be clear in the text whether you have accounted for isoprene fragmentation (mostly to m/z 41) because as you probably know isoprene sensitivity is significantly deviating from transmission estimate vs non-fragmenting species (e.g. MVK). Not accounting for fragmentation would result in underestimating the concentrations but perhaps you derived a fragmentation correction factor for proton reaction rate constant (effective k) in the post-campaign calibration? In either case it is not clear so you should add appropriate detail to SI.

17) Sect. 3.2 Figure 2a,c is incredibly super cool, and the diurnal emission potential seems relatively constant as expected, except for the morning and evening times. Did you try to filter for low u-star to see how this would affect the diel trend? Maybe you could plot the low ustar data in grey. Do you know why you could not reproduce this stability with G93 as beautifully as with G12?

18) Figure 7. This is also an incredibly beautiful figure. In particular the temperature activity factor works shockingly well. In panel a, it might also be useful to add the parameterized G06 line which would better fit the gamma for PAR. It would be nice to further discuss these differences because they show major results from this study.

[Figure]

**Technical:**

19) G93, G95, G06, G12 need to be defined on their first use and used consistently (e.g. G93 in the abstract).

20) add page numbers in SI

21) Sect. 2.1.1-2.1.5 Significant figures in the coordinates of the locations vary from 3 to 10. Please be consistent.

22) P6 L13 the unit of the gas constant seems incorrect. Maybe a typo or maybe you intended to refer to 1 mole.

**References:**

Langford, B., Misztal, P. K., Nemitz, E., Davison, B., Helfter, C., Pugh, T. A. M., MacKenzie, A. R., Lim, S. F., and Hewitt, C. N.: Fluxes and concentrations of volatile organic compounds from a South-East Asian tropical rainforest, Atmos. Chem. Phys., 10, 8391-8412, https://doi.org/10.5194/acp-10-8391-2010, 2010.

---

## Referee Comment (RC3) · Anonymous Referee #3 · 25 Jul 2017

Overall, this is a nice paper that explores a technical aspect of isoprene emission modeling: relating whole-system, measured isoprene fluxes to the emissions capacity used in most isoprene emission frameworks. My biggest concern is that the authors recommend using the means of observations and of the calculated gamma to find the emission capacity (equation 6). On page 13, line 2, the authors' state that the superiority of this technique has been established in the previous results section. Since the least-squares approach has a well-established theoretical justification, the manuscript should do more to explore the advantages of Equation 6. This must be a pretty common

issue in modelling. For example, how do ecosystem models of net primary productivity deal with this issue? I think the authors could do more to justify this new approach.

Major comments

*Figures 2 and 3 pack in too much information. For example, I was interested in comparing the performance of the LSR & ODR approaches with MEGAN. In most cases in Figure 3, I could not distinguish these two cases because of overlapping plotting characters. What's the benefit of plotting all the different time average periods? Couldn't that be conveyed in a separate graph? Near lines 28-31 on page 12, you take away from Figure 3 that the G93 approach difference significantly from the MEGAN approach. This is well known, and could be conveyed more succinctly in a separate figure.

*The conclusion that "the emission potential is not constant throughout the day" should be refined. Within the modeling framework, the emission potential should be a constant throughout the day. The better way to frame this is that the calculated emission potential is not properly capturing the diurnal cycle. Also, considering just 08:00 to 18:00, there's not much variation in the EIP.

*On lines 9-12, page 8, you mention the issue of the intercept for the least-squares approach. For the least-squares calculations in this paper, did you use a zero intercept?

Minor comments

*The abstract is a bit long. While comprehensive, I counted 660 words. In particular, some of the recommendations at the end repeat material from the abstract (factor of four). A target of 600 words seems more reasonable. With an open-access journal, there is less pressure on fitting so much in the abstract.

Page 2, lines 33-34: The article by Arneth would be useful to consider and site at this point in the discussion (http://www.atmos-chem-phys.net/8/4605/2008/).

Page 2, lines 34-36: Very minor point: branch enclosure measurements typically can't

be performed at standard conditions. Instead, leaf temperature and light are measured, and often the Guenther algorithms are applied to derive a basal rate.

Page 3, lines 5-7: Again, a good place to refer to Arneth et al 2008.

Page 3, line 22: Inconsistencies isn't the right notion here. Yes, there are inconsistencies, but there are also different assumptions.

Page 4, line 1: Since the algorithms for previous light and temperature are coming to come into play, some mention of the meteorological conditions during the campaigns compared to average climatology is necessary. In particular, where any of the campaigns conducted during times of water stress?

Page 7, lines 21-32: This is a lot of text to describe something that wasn't used. Please consider if its necessary to include.

Page 8, line 25: Shouldn't this produce the same result as a linear regression with the intercept set to 0?

Page 11, lines 16-18: "discernable" is subjective. This might be a real effect, or it might be random noise. Also, connect this to the major comment above: this variation represents a failure in the underlying model. Lines 26-27 (page 11) are the proper way to frame this conclusion.

Page 11, lines 40-41: This is a major drawback to the ODR approach. Good, you reach this conclusion lines 3-5 on the following page.

Page 12, lines 36-41: Yes, but this is only true when considering the extreme ends of the day. Typically, the focus is 10:00 – 16:00, when the variability is much lower with MEGAN.

Technical comments

Page 1, line 34: hyphenate 'site specific'

[Figure]

Page 2, line 18: hyphenate 'ground level'

Page 3, line 39: note explicitly this is ug of isoprene, not carbon (ugC), which has also been used in the past.

Page 4, line 13: be consistent about lat/long significant figures. The two used elsewhere are probably sufficient.

Page 4, line 23: According to BG style, "32m platform".

Page 7, line 7: hyphenate "in canopy"

Page 10, lines 6-7: fix grammar

Page 10, line 13: reflect should be reflects

---

## Author Comment (AC2) · 22 Aug 2017

Our response to each reviewer are compiled in our response to reviewer 1.

---

## Author Comment (AC3) · 22 Aug 2017

Our response to each reviewer are compiled in our response to reviewer 1.

---

## Author Response (AR1)

Dear Georg,

In this document you will find a brief summary of the main changes made to the manuscript / supplementary information, our response to each of the reviewer's comments and a revised version of the manuscript and SI with track changes highlighted.

Best Wishes,

Ben.
* * *
**Summary of main revisions**

Reporting Fluxes for defined conditions

Alex Guenther's made an excellent suggestion to include a further approach that calculates the emission potential by reporting the average flux for a set of defined (e.g. not standard) conditions. In response to this we have introduced this approach to Section 2.3 where we outline the various options available. We have then added a further section (3.4 – reporting fluxes for a set of defined conditions) to the results section where we discuss our approach to determine the most appropriate set of defined conditions to use (See also Supplementary Information, Section S7, Figs S11-S14) and the advantages and disadvantages of this approach relative to the other methods discussed in the paper. Within this section we include an additional table (4) and figures (8 and 9).

Additional Recommendations

Alex Guenther made several suggestions for further recommendations which we were keen to include in the revised manuscript. These relate to (i) Leaf-level experimentalists reporting emission potentials on both a per mass and per area basis to avoid unnecessary uncertainty when comparing with above canopy fluxes (see Section 3.5) (ii) Modellers providing a single point version of their code to allow experimentalists to more easily translate their above canopy fluxes to isoprene emission potentials (see Section 4) (iii) Experimentalists to no longer use the G06 model in a big leaf format because the resulting emission potentials are typically biased low (see Section 3.3) and (iv) that emission potentials calculated using the weighted average method are also reported for a set of defined conditions to allow modellers to work back to standard conditions themselves (see Section 3.4).

Correction of Castelporziano data

The reviewers noticed that the PPFD data reported for the Castelporziano site looked high. We have checked this with the data provider and the reviewers were correct, the data had been labelled with the incorrect units. We have now reanalysed the Castelporziano data using the correct PPFD data. This has resulted in changes to the Castelporziano results shown in Figs 1, 3, 4, 5, 6 and 7 and discussion of the Castelporziano results made

throughout the text. We have also revised all of the emission potentials and figures presented for the Castelporziano site in the Supplementary Information.

The use of the Weighted Average Method

Two of the reviewers had queries about our decision to recommend the weighted average method over the LSR approach. We discussed our reasoning in detail in our response to the reviewers and have now made changes to the manuscript to make it clear that (i) our reasoning for choosing the weighted average is taken from the perspective of accounting (e.g. we want to ensure the algorithm can reliably reproduce the average flux) and (ii) our reasoning for not adopting the LSR technique is that the regression between flux and gamma is typically non-linear (due to the algorithms inability to properly represent the attenuation of light and temperature through the canopy) and hence violates the fundamental assumptions of a LSR approach (see Section 2.3).

Species composition uncertainties

The reviewers appreciated the fact that we had made an attempt to properly quantify all of the uncertainties associated with derived emission potentials, but questioned if looking at emission potentials by wind sector might give a better approximation of the uncertainty in species composition than assigning a somewhat arbitrary value of 10%. In response to this I have looked at wind roses of emission potential at the Ispra, Bosco fontana and Alice Holt sites and now use the variability between wind sectors as an estimate for species uncertainty (see Section 2.7 and the Supplementary Information (section S6, figure S10). Because the Castelporziano and O3HP sites had much shorter time series it was not possible to perform reliable wind rose analysis at these sites so the uncertainty in species composition was increased to 20% at these sites to reflect the higher values observed at the three other sites.

**We thank Alex Guenther and two anonymous referees for their valuable comments on our manuscript. The general feeling seems to be that this paper will form a useful part of the scientific literature but could be further strengthened through the inclusion of emission potentials based on measurements made near a set of defined conditions and through the addition of further recommendations of best-practice for both above-canopy and leaf-level experimentalists. We are, of course, happy to address both of these recommendations in our revised manuscript. In addition we address each of your individual concerns in our more detailed response below.**

**Response to Reviewer 1, Alex Guenther**

General comments

1) The most important comment that I have is that the authors should consider an approach that recognizes that measurements similar to the standard conditions should be used to determine the emission potential. I recognize that this may not result in the best estimate of the daily total emission or the emission at conditions dissimilar to the standard conditions but I would argue that is not the "job" of the emission potential. The emission potential, by definition, is the emission at specified standard conditions and so the best estimate is made by either selecting measurements within some fairly narrow range of conditions or perhaps weighting measurements by how close they are to the standard conditions. It is the "job" of the emission algorithm to go from the emission potential at standard conditions to the emission at other conditions. If the emission algorithm does not do a good job of this then there will be errors but you shouldn't bias the emission potential to try account for this. Instead you should work on developing a better emission algorithm. There will be relatively little difference in the emission potential calculated by different emission algorithms if the emission potential is based on measurements made under conditions similar to the standard conditions. Of course, the problem with applying this approach to canopy scale flux data is that we can't control the measurement conditions and there may not be any that are similar to the standard conditions. If that is the case then the emission potential should be reported for some standard condition that is within the range of the observed conditions. You could then leave it up to the developers/users of a given model to convert this to an emission potential for the standard conditions of their model. Looking at Figure 1, it appears that 3 of the sites would have some measurements at T=30, PPFD=1500 while T=25, PPFD=1500 might be appropriate for the other two. You could report the emission potential for T=25 as the measured emission potential and then also report one or more calculated emission potentials for T=30 along with an exact specification of the model approach used to get there.

**AR: In its current form our manuscript is written very much from a measurement perspective so we certainly welcome the opportunity to learn what those involved in BVOC emission model development would find most useful in order to ensure that the recommendations we make are consistent with the needs of the community. We fully agree that if the algorithms were perfectly describing the response to the environmental conditions, the problem our paper discusses would not exist. Thus, developing a better algorithm is a great solution, but until a perfect algorithm is developed the problems we outline will persist.**

We also agree with your assertion that each algorithm would derive a similar emission potential when working with only values based close to standard conditions, because the γ factors that account for deviation from the standard conditions would all be small. However, as formulated the standard conditions of most models are far removed from conditions found, e.g., in higher latitudes including at some of our sites. Thus, even if the measurements are made at the reference conditions under laboratory conditions there are two problems: (a) the plant species may not be adapted to these reference conditions and the emissions may not be representative and (b) the algorithm then still needs to correctly extrapolate from those reference conditions to the modelled conditions. The only problem that is solved by this approach is that a different algorithm may otherwise be used in extrapolating from the measurement conditions to the reference conditions than is used for extrapolating from the reference to the modelled conditions, thus duplicating the uncertainty.

Nonetheless, we like your suggestion in principle to report an emission potential together with the conditions that is typical for the measurement dataset and then leave it to the emission modeller to infer the emission potential at their model's reference conditions that is consistent with these emissions under given conditions.

However, there are two problems. As we mention in the manuscript, our reasoning for not including an emission potential based on measurements made close to standard conditions is that the percentage of data that meet these criteria is incredibly low. In addition, as emission algorithms are getting increasingly complex, this is reflected in an increasing number of reference parameters, which means the space of measurement conditions has to be stratified in many dimensions. We had therefore not considered reporting an emission potential for a different set of defined conditions but are happy to do so in the revised manuscript and to include this as a recommendation if you are think this would be useful for the modelling community.

Focussing only on the instantaneous responses to PAR and temperature, we have subsequently re-analysed each data set using a 2d histogram to identify the most appropriate set of conditions to use, e.g a period near the solar maximum with sufficient frequency to provide a robust average flux. We searched for fluxes within windows of ± 0.5 K and ± 100 µmol m$^{-2}$ s$^{-1}$ PPFD. Figure 1 shows the results for the Ispra forest site, which highlights the most abundant set of conditions to use to be between 302-303 K and 1600-1800 µmol m$^{-2}$ s$^{-1}$ PPFD, yielding a total of 19 flux measurements to average. We limit our defined conditions to just current light and temperature as refining the search further to account for the other gamma terms (e.g. T24, T240, PPFD24, PPFD240, RH, wind speed) would limit the available data to little more than n=1. Instead, in Table 1 we list the average fluxes for the defined conditions along with the average of the gamma terms with associated standard deviations.

[Figure]

**Figure 1. 2d histogram showing the number of flux measurement made at the Ispra forest site that fall within defined light (± 100 PPFD) and temperature (±0.5 K) bins.**

As you suggest, reporting fluxes for a set of defined conditions in this way will allow model developers to convert these to the standard conditions used in their model, but, as you point out, will unavoidably introduce further uncertainty. To investigate this further we derived new emission potentials "converted" from the measured values in Table 1 for the G93 and MEGAN 2.1 models (a, b and c) and then compared the performance of these algorithms at replicating our measured fluxes at each site. Figure 2 shows the percentage difference between the averaged measured flux and the averaged modelled flux when using the "converted" isoprene emission potentials. The calculated bias ranged between +29% and -4% for the G93 algorithm and between +9% and -40% for the MEGAN 2.1 approaches. The bias for the G93 algorithm is typically positive which reflects the fact that the algorithm performs well at conditions close to standard conditions but performs worse in the morning and afternoon, overestimating emission fluxes due to its inability to account for the attenuation of light and temperature through the canopy. The observed bias in the MEGAN2.1 simulated isoprene fluxes is driven by two factors (i) the fact that the average flux for the set of defined conditions is based on a limited number of data points (which affects both algorithms), ranging between n=4 to n=19, which may be a poor representation of the typical flux footprint and canopy heterogeneity and (ii) the defined conditions are based on current PPFD and temperature with larger uncertainty on the remaining gamma terms such as past PPFD and temperature.

Thus we conclude that this approach, by definition, succeeds in simulating the emissions at 'typical' conditions encountered at the site, but not in reproducing the average emission.

**Table 1. Isoprene emission potentials for each of the five sites for a set of defined conditions. Numbers in brackets show 1 σ.**

| | Alice Holt | Bosco Fontana | Castelporziano | Ispra | O3HP |
|---|---|---|---|---|---|
| IEP (average flux) [μg m$^{-2}$ h$^{-1}$] | 2143 | 1911 | 83 | 9404 | 2649 |
| σ [μg m$^{-2}$ s$^{-1}$] | 1075 | 599 | 102 | 3593 | 988 |
| $\overline{RE}$ [μg m$^{-2}$ h$^{-1}$] | 142 | 443 | 31 | 1268 | 353 |
| $N$ [#] | 9 | 17 | 5 | 19 | 4 |
| Temperature range [K] | 293-294 | 302-303 | 300-301 | 302-303 | 293 - 294 |
| PPFD range [μmol m$^{-2}$ s$^{-1}$] | 800-1000 | 1800-2000 | 1400-1600 | 1600-1800 | 1800-2000 |
| Mean Temperature [K] | 293.4 (0.2) | 302.5 (0.3) | 300.5 (0.14) | 302.6 (0.3) | 293.7 (0.16) |
| Mean PPFD [μmol m$^{-2}$ s$^{-1}$] | 915 (66) | 1902 (60) | 1523 (44) | 1703 (61) | 1852 (35) |
| Mean 24 T [K] | 290 (1.1) | 299 (1.4) | 295 (0.6) | 298 (1.6) | 290 (0.9) |
| Mean 240T[K] | 290 (0.94) | 299 (1.8) | 295 (0.25) | 297 (1.4) | 290 (1) |
| Mean 24 PPFD [μmol m$^{-2}$ s$^{-1}$] | 432 (84) | 680 (70) | 424 (31) | 556 (3) | 625 (54) |
| Mean 240 PPFD [μmol m$^{-2}$ s$^{-1}$] | 415 (92) | 659 (48) | 452 (15) | 553 (17) | 591 (0.7) |
| Humidity [g/kg] | 7.9 (1.2) | 11.9 (1.6) | 13.5 (1) | 11.4 (1.7) | 6.5 (0.8) |
| Wind Speed [m s$^{-1}$] | 2.19 (1) | 2 (0.81) | 1.8 (0.5) | 1.4 (0.5) | 4.1 (1.4) |

As we discuss in the original manuscript, if emission potentials are calculated using all measured flux data and not just those obtained at a set of defined conditions, then the average measured flux can be replicated by the algorithm with zero bias assuming the weighted average approach has been used to derive the emission potential. The drawback to this approach is that that emission potential cannot then be easily converted for use in different emission models. We agree with you that publishing an average flux for a set of defined conditions may be more readily used by model developers and hence have a wider impact, but we believe it is necessary to highlight that this approach results in emission potentials that are inherently more uncertain, especially for the more complex algorithms where not all of the gamma terms can be controlled. In the revised manuscript we will have a further section to discuss the findings shown here and will recommend experimentalists to adopt both approaches. In addition, we will further stress the importance of researchers submitting their observational data sets to online, publically accessible, data repositories such as the VOCsNET database, as we believe a well populated community database would be a far more valuable resource to model developers and would support further improvement in emission algorithms.

[Figure]

**Figure 2. Percentage bias of the average isoprene emission flux simulated by the G93 and MEGAN2.1 emission algorithms at the five measurement sites, Alice Holt (AH), Bosco Fontana (BF), Castelporziano (CP), Ispra forest (ISPRA) and the Observatoire de Haute Provence (O3HP), compared to the measured average flux when using a "converted" emission potential.**

2) I assume that there is at least some landscape heterogeneity at some of these sites. The authors should consider binning measurements for different "footprints" associated with different wind directions that represent different oak fractions. This could provide "replicates" with emission potentials for a larger range of oak fraction values that may provide some insights into the value and variability in the leaf-level emission potential. Of course, this assumes that there is some information on the landscape heterogeneity at these sites.

**AR: We agree that such an approach could prove useful where there is very detailed species composition data available. Unfortunately, the information we have on species composition at each site is for the forest as a whole and not spatially resolved. Nonetheless, calculating an emission potential by wind sector does provide some information on the spatial variability of the emission potential. For each site we will investigate to see if there is sufficient variability in the wind direction to enable us to infer a species composition uncertainty based on the variability of emission potentials calculated for each wind sector. We have explored the spatial aspects of species composition for the Bosco Fontana field site in a separate paper (Acton et al., 2016) and will refer to the main messages in the revised manuscript.**

Specific comments

Page 2, line 28: "In the Guenther algorithms, isoprene emission rates are modelled by assessing the emission potential". This is not something specific to these algorithms all isoprene emission models include some term of this type, although they may not call it an emission factor.

**AR: We will change this to " In most BVOC emission algorithms…"**

Page 4, line 31: delete "to" in "Castelporziano has to a Thermo-Mediterranean"

**AR: This will be changed**

Page 6, line 18 and line 33: Be more specific about how the tendencies for studies to use a big leaf approach and using leaf temperature equals air temperature. For example, how many of the studies listed in line 16/17 do this? It may be useful to consider that at least one reason investigators do is because of the considerable effort involved in applying the full inverse canopy algorithm to their dataset and it would be useful to have an easier way to do this. For example, Yu et al. al 2017 (http://doi.org/10.1016/j.scitotenv.2017.03.262) calculated emission factors using an aircraft flux measurement dataset by using the single point version of MEGAN2.1 that you mention on page 7 line 9, and is relatively easy to use, and compare this with emission factors estimated using the regional MEGAN2.1 FORTRAN code, which is relatively difficult to use. A possible recommendation from your study is that BVOC emission modelers should provide a single point version of their code that can more easily be used to derive emission potentials from tower and aircraft flux data.

**AR: We agree wholeheartedly with you on this point. The big leaf G93 approach is undoubtedly the most widely used method to calculate emission potentials due to its simplicity. Our investigation of different algorithms was only made possible through the use of the "Pocket MEGAN" you provided so we will ensure the revised manuscript includes a recommendation for model developers to provide a single point version of their code to enable experimentalists to more easily calculate emission potentials.**

Page 8, line 32-36: The statement that "Measurements of the emission potential made using leaf-cuvette systems on a single leaf or branch gives a direct measurement of the isoprene emission rate that inherently excludes the deposition process." seems inconsistent with "but it may still be offset slightly as some of the isoprene may undergo dry deposition to leaf surfaces". The leaf cuvette measurement excludes deposition to other leaves and to the soil but there is the possibility of uptake by the emitting leaf including by phyllosphere microbes on the leaves.

**AR: You are correct. We will amend the text accordingly**

Page 10, line 27: what is the basis for the 10% uncertainty assigned to species composition and 15% to LAI? Does this consider landscape heterogeneities and the uncertainty associated with differences in the LAI and species composition within the footprint of each measurement in comparison to the average for the whole area?

**AR: The species composition data and information we have on LAI for each measurement site did not come with associated uncertainties and therefore these values are fairly arbitrary. As discussed above, we will revise the 10% species composition based on the spatial variation of isoprene emission potentials when broken down by wind sector. An initial analysis of IEP windroses for the AH, BF and Ispra forest sites and shown in Figure 3, reveals that the emission potential is fairly constant with wind direction. Taking the standard deviation of the IEP from different wind sectors and comparing with the site average suggests a variability of 14% to 28%. The largest variability was seen at the BF site (28%), which had the smallest fraction of oak and the smallest was seen at AH (14%) which was composed of 90% oak. Wind rose analysis were not performed on the two remaining sites, Castelporziano and O3HP, because these were much shorter time series with insufficient data coverage to provide meaningful emission potentials for different wind sectors. In the revised manuscript we will increase the uncertainty for these two sites from 10% to 20%.**

[Figure]

Figure 3. Isoprene emission potentials calculated by wind sector for the Alice Holt (a), Ispra forest (b) and Bosco Fontana (c) sites (red traces) compared to the site average emission potential (blue trace).

Page 10, line 34: The specific leaf mass that you use to convert canopy scale emission potentials to leaf-per-mass emission potentials are arguably as uncertain and variable as isoprene emission potentials. A 25% uncertainty for specific leaf mass may be a reasonable value but you should justify this number and mention how this makes it difficult to compare canopy and leaf scale measurements. This uncertainty could be eliminated if the investigators making leaf level measurements would report emissions as both "per mass" and "per area" leaf emission potentials (i.e., they should provide the specific leaf mass for each measurement) and I suggest that this be a recommendation. If some of the leaf level data that you refer to include measured specific leaf mass (and so direct measurements of per-area leaf emission potentials) then you should make this more direct comparison that does not suffer from the large uncertainties in specific leaf mass estimates (you could do this in addition to the comparison you have already made with the per-mass leaf emission potentials.

**AR: This is an important point which we will add to our discussion along with specific recommendation for leaf-level emission potentials to be reported on both a "per mass" and "per area" basis.**

Page 11, line 29: Define what you mean by a "wide" range. The range given here of 6750 +/- 1150 is equivalent to +/- 17% which is well within the uncertainties that you discuss. Should that be considered a wide range?

**AR: This will be changed to "…the calculated emission potentials span from ~ 5,600 to 7,900 µg m-2 h -1"**

Page 12, line 22: "regional or VOC global" should be "regional or global VOC" Page 13, line 26: MEGAN2.1 allows users the option of using a constant value for each of the 15 PFTs but the recommended approach is to use the MEGAN2.1 isoprene emission factor map that accounts for the fraction of isoprene emitters in each landscape based on plant species composition and the species specific emission potential for each location.

**AR: We will make the suggested change and highlight the suggested MEGAN best practice in the revised manuscript.**

Page 14, line 1: The MEGAN2.1 canopy-scale emission potential for high isoprene emitters is 24000 ug m-2 h-1. The global average temperate broadleaf deciduous tree PFT isoprene emission potential of 10000 thus represents a canopy composed of 41.6% high isoprene emitting trees which is high but not "primarily composed" as stated in the text.

**AR: We will make this point clear in the revised manuscript and rephrase our statement accordingly**

Page 14, line 17-22: As is pointed out in this manuscript, canopy-average leaf-level PPFD values are considerably lower than above canopy values. Even sun leaves have a PPFD that is typically 50% or less than the above canopy PPFD since they are, on average, at an angle to the sun. The MEGAN2.1 standard condition for the past 24 and 240 hour PPFD refers to the leaf-level value and it is not appropriate (i.e., it just doesn't work) to use the above-canopy value (i.e., a big-leaf model) with this equation (G06). That the G06 past 24/240 hour algorithm should

not be used with the big leaf model is an important point to make in this paper but then going on to compare the MEGAN leaf-level PPFD standard condition with the measured above-canopy PPFD in figures S5 to S9 is comparing "apples and oranges" and may be confusing to some readers. It should be made clear that this is a comparison of two different parameters (above canopy PPFD and leaf-level canopy-average PPFD) and the main point is that the above-canopy value should not be used in the past 24/240 algorithm.

**AR: Agreed. We will make this point clear in figures S5 to S9.**

Page 14, line 28: Why not just conclude/recommend that the G06 algorithm should not be used with a big leaf model?

**AR: Agreed. We will add this recommendation here.**

Page 14, line 23 (and Figure 1 and Figure 7): Check on the values of PPFD shown for Castelporziano. They appear to be higher than what would be expected at the top to the atmosphere. Also, note that PAR should always be expressed in units of W/m2 while PPFD is the appropriate term when you use units of micromol/m2/s.

**AR: Agreed. We checked with the data owner who has now provided the PAR data in the correct units. We have re-analysed all of the Castelporziano data using the correct PAR data and have updated the text, tables and figures accordingly. Additionally we have now replaced PAR with PPFD throughout the manuscript.**

Page 14, line 37: This sentence is confusing.

**AR: This will be changed to "Interestingly, when the use of previous light and temperature is switched off (e.g. MEGAN 2.1 (c)) the emission potential increases as the effects of past light and temperature are no longer considered.**

Page 14, line 40: This may be because the Castelporziano PPFD solar radiation value is incorrect as mentioned above.

**AR: Thank you for pointing out this error. We now have the correct data from the data provider and have recalculated all of the emission potentials and replotted all graphs and tables to account for the adjusted PAR values.**

Page 15, line 4-6: or when they are measured under conditions similar to the standard conditions

**AR: Agreed, we will stress this point in the revised manuscript.**

Page 15, line 9: Leaf-level isoprene emission potential varies considerably between the top and bottom of the canopy and also depending on the past light and temperature environment. Are the leaf level emissions representative of the canopy average, as is the case for the canopy scale measurements, and is the past light and temperature similar? If this is not known, and it is often not reported for leaf-level studies, then this point could be included in the discussion of uncertainties for this comparison.

**AR: Agreed we will add this point to our discussion and make a recommendation for past light and temperature to be reported with Leaf-Level emission potentials.**

Page 15, line 23: As discussed above, an alternative approach is to select only measurements that are close to the standard conditions.

**AR: This will be added.**

Page 16, line 37: This is an important point and a good opportunity for you to provide some recommendations for the standardization of flux measurements.

**AR: Standardisation of VOC flux measurements is undoubtedly important, but we are not comfortable with making specific recommendations without fully engaging with the community. Encouragingly, some progress in this area is being made with a PTR-MS intercomparison scheduled for later this year in Cabauw as part of the European research infrastructure project ACTRIS. This will hopefully lead to the formation of standard instrument operating procedures but a similar effort is needed for flux measurements and in particular for their post-processing.**

Figures 4-6: You were generally consistent in referring to "emission potentials" but these figures refer to "emission factors". Either can be used but be consistent.

**AR: These will be changed**

**Response to Anonymous Reviewer 2**

General

1) The authors should try to avoid the confusion between the same parameters derived in a different way/scale/conditions. Alex's point to use the conditions closest to the standard conditions seems like another sensible approach worth evaluating. However, inverting the algorithm even at conditions significantly deviating from standard conditions seems still worth the exercise but must necessarily lead to larger errors from environmental parameters measured simultaneously, and potentially may become inconsistent with original model design or intent. Assuming the measured environmental parameters (e.g. T, PAR) are accurate, the value of inferring about the emission potential at different conditions seems valuable to assess how well the algorithmic activity factor works. If it works well, then the emission potential collected under the conditions close to standard should be similar to that inverted from fluxes measured at different conditions with reverse algorithm within the

same footprint. For example, Figure 2 showing stable measured emission potential during the day is unbelievably encouraging, so this approach in my opinion deserves some greater attention.

**AR: We agree that Alex's suggestion is a good one and have added emission potentials for specific light and temperature conditions which we discuss further in our response to Alex and will also include in the revised manuscript.**

2) Model parameters which were designed for the leaf level-scale may not always be compatible for comparison or extrapolation with the same parameter obtained from inverting the equation at the ecosystem scale, even if in principle it should work. For this reason, it would be helpful to use a thoughtful system of descriptors for equivalent parameters, e.g. EFcan or EFextrap, so it is clear and distinguishable how the parameter was obtained. This will help the issue which the authors are trying to communicate to modelers (last paragraph in the abstract) that they should be careful about how these parameters were derived before using them.

**AR: In the supplementary information we do already make this distinction by presenting emission potentials as $E_{eco}$ (ecosystem emission potential), $E_{can}$ (Oak canopy emission potential) and $E_{LL}$ (Leaf-level equivalent emission potential). We will add the subscripts to figures 4-6 and throughout the text.**

3) The abstract seems somewhat heavy for a reader. The take-home message about the differences as large as a factor of four are somewhat scary and confusing. It asks for some further insight as to what exactly causes such a large difference. If you suggest the uncertainties in the inversion of the algorithms are different for different models is it because the inversion does not work perfectly or the specific algorithm does not work well for top-down inference about the emission potential (so would likely not be accurate the other way round – bottom up)? I suggest to focus in the abstract on the major points and progress, and less so much on what you did and technical detail. Specific

**AR: Agreed, the abstract will be made more concise with less focus on the technical detail.**

4) P5 L33 G93 "Perhaps the most widely used" – did you mean the most highly cited?

**AR: I think it is the most highly cited because it is also the most used. We will change to say "…the most widely used and highly cited…"**

5) P8 L4-5 Why did you leave out Langford et al. 2010 here? Misztal et al. (e.g. 2011, table 3) used approaches to estimate BER from the regression with measurements, as well as from the middle of the day (11:00 LT; which you show also here agrees well). I think you should add Langford et al., 2010 reference here, because they reported BERs as mid-day average. I would also suggest to be more neutral and refrain from subjective statements about which approaches are more popular.

**AR: This reference will be added**

6) Abstract. Seems long and overloaded. In particular the last two sentences are rather pessimistic and might agitate modelers unnecessarily, because it is hard to believe you could really be off by a factor of four if everything is done perfectly or at least it is not sufficiently clear why exactly this is the case.

**AR: The last two sentences will be removed.**

7) In the concluding remarks, you focus on the way the emission potentials are derived. Do you also want to make a bigger point about how the future models could be enhanced to better assimilate observational data at regional scale?

**AR: We will stress the point that by providing a consistent and robust approach to calculating emission potentials from top-down flux measurements future models may be better parameterised through the incorporation of regional scale observations.**

8) It is great that you include the original definition of emission factor (collected under the standard conditions and leaf scale). I wonder if it would be worth making a distinction between the parameterized algorithm on the full-canopy observations and whether it should be labeled as the same or a modified algorithm.

**AR: We are unsure to which part of the manuscript you are specifically referring to here. Please could you provide a specific page and line number?**

9) Table 1 – since PTR-TOF-MS was used in Castelporziano, why did you write vDEC? Did you artificially convert the PTRTOF dataset to disjunct to be consistent with other measurements? Either seems fine, as long as it is clear.

**AR: This has been changed**

10) SI S1.1 Alice Holt – Measurement setup Lag time - as the signal to noise ratio for isoprene was rather very high, why did you use the approach for low signal to noise species? Why did you not use the accurate lag-time from each half hour period?

**AR: The signal-to-noise ratio for the isoprene data set was well below 10. According to Langford et al. (2015) a data set with a signal-to-noise ratio in this range and with disjunct sampling interval of 2.5 s could expect a systematic bias of around 50% (see Figures 4 and 6b). For this reason we used a prescribed lag-time as recommended by Langford et al.**

11) SI S1.1 "to ensure the reduced electric field strength" seems somewhat random and out of context. Also 2.01 mbar suggests that the pressure was stable to 0.01 mbar. This is rarely the case. I suggest you say 2.0 mbar or 2.01+/-0.XX mbar

**AR: The line you refer to simply describes that the E/N ratio was held constant at 127 Td. The E/N ratio is a fundamental parameter which should always be reported so we would be reluctant to remove this. "The PTR-MS operating conditions were held constant throughout the measurement period to ensure the reduced electric field strength (E/N, where E is the electric field strength and N is the buffer gas density) was maintained at 127 Td."**

12) SI S.1.1 P.1 L21-22 Instead of the justification it might be appropriate just to write what the consequences are (if any). I do not think it is necessarily bad to use high resolution measurement if it is appropriately post-averaged unless it leads to counting zeros. Otherwise, can you inform what the difference is between fluxes measured at 50 ms and averaged to 200 ms, as opposed to measured at 200 ms?

**AR: The consequences are a lower signal-to-noise-ratio and potential systematic bias. We avoid this potential bias by using a prescribed time-lag as recommended by Langford et al. (2015). We will make this point in the revised manuscript.**

13) P8 L16-30. Unfortunately, I am extremely confused by the lack of clarity here. In particular, the weighted IEP is concerning. Why do you average the activity factor across the footprints and conditions before taking the ratio? It does not seem appropriate, because, as you say, these processes are nonlinear. For example, you have to use the model to average PAR accurately. It is more intuitive to average the emission potential, because in principle it should be relatively constant for the same vegetation type (as you show in Fig. 2), and you would not have to average nonlinear processes.

**AR: We apologise for the lack of clarity as we believe the reviewer may have misunderstood our approach. We are not calculating a gamma for the average meteorological conditions, but calculating the average of all gammas which were explicitly calculated for each individual flux period. Please also refer to our response to reviewer 3 where we further justify this approach. In the revised manuscript we will clarify our approach to avoid any further confusion.**

14) Sect. 2.4. Isoprene deposition. Given the large gradient it is interesting that the authors suggest the deposition can be significant even for isoprene. It would be helpful to provide the percentage range of isoprene deposition relative to total flux, in addition to canopy resistance. As Alex wisely points out, you need to be aware of epiphytic microbes whose role is not yet well understood in affecting emission and uptake of isoprene.

**AR: This is already stated (5-8%) in both the abstract and results sections. As we point out in the manuscript these estimates are highly dependent on the value of Rc we use, which may not be ideal for our sites but represents the only published value available in the literature. To truly understand how much isoprene is lost due to dry deposition and indeed to microbes on leaf surfaces will require further research,**

**but the method we outline will become increasingly meaningful as more VOC specific canopy resistances become available in the future.**

15) Sect. 2.6 how do you differentiate between the effective LAI and the tree cover area fraction?

**AR: Unfortunately we do not have leaf area index measurements for the individual tree species, only the tree cover fraction and hence we cannot differentiate between the two. Upscaling to 100% oak undoubtedly means that changes to the canopy LAI will occur, with the largest changes associated with sites with the lowest fractions of oak. We discuss this uncertainty in Section 2.7 and attempt to scale this relative to how much upscaling is required but recognise that without detailed information on tree species LAI our efforts are somewhat arbitrary.**

16) P10 L28 As you did not calibrate isoprene on gas standard at Alice Holt, you had to estimate the concentration from relative transmission. I am generally fine with the approach, but it should be clear in the text whether you have accounted for isoprene fragmentation (mostly to m/z 41) because as you probably know isoprene sensitivity is significantly deviating from transmission estimate vs non-fragmenting species (e.g. MVK). Not accounting for fragmentation would result in underestimating the concentrations but perhaps you derived a fragmentation correction factor for proton reaction rate constant (effective k) in the post-campaign calibration? In either case it is not clear so you should add appropriate detail to SI.

**AR: The reviewer is correct about the fragmentation of isoprene to m/z 41 and this was already accounted for in the reaction rate constant k used in our transmission. For completeness, we now include a description of the correction applied.**

17) Sect. 3.2 Figure 2a,c is incredibly super cool, and the diurnal emission potential seems relatively constant as expected, except for the morning and evening times. Did you try to filter for low u-star to see how this would affect the diel trend? Maybe you could plot the low ustar data in grey. Do you know why you could not reproduce this stability with G93 as beautifully as with G12?

**AR: We are glad the reviewer enjoyed this figure. The fact that the G12 algorithm is able to much better replicate the measured fluxes, even during the evening and morning periods means that low turbulence is not the reasoning for the comparatively poor performance of the G93 algorithm. The G93 is unable to replicate the diurnal pattern because it uses the big-leaf approach and therefore cannot properly capture the effects of light and temperature attenuation through the canopy.**

18) Figure 7. This is also an incredibly beautiful figure. In particular the temperature activity factor works shockingly well. In panel a, it might also be useful to add the parameterized G06 line which would better fit the gamma for PAR. It would be nice to further discuss these differences because they show major results from this study

**AR: The purpose of this figure is to demonstrate why the G06 algorithm generates emission potentials that are much lower than the other algorithms. We feel it is critical to highlight the problems with this approach because it is becoming more widely used, including by ourselves in the past. We feel that adding the PCEEA approach to this figure, an algorithm that was more consistent with the emission potentials calculated using MEGAN 2.1, may dilute our message.**

Technical:

19) G93, G95, G06, G12 need to be defined on their first use and used consistently (e.g. G93 in the abstract).

**AR: This will be changed**

20) add page numbers in SI

**AR: Page numbers will be added**

21) Sect. 2.1.1-2.1.5 Significant figures in the coordinates of the locations vary from 3 to 10. Please be consistent.

**AR: Changed to 3 SF at each site**

22) P6 L13 the unit of the gas constant seems incorrect. Maybe a typo or maybe you intended to refer to 1 mole.

**AR: This will be corrected.**

**Response to Anonymous Reviewer 3**

Overall, this is a nice paper that explores a technical aspect of isoprene emission modeling: relating whole-system, measured isoprene fluxes to the emissions capacity used in most isoprene emission frameworks. My biggest concern is that the authors recommend using the means of observations and of the calculated gamma to find the emission capacity (equation 6). On page 13, line 2, the authors' state that the superiority of this technique has been established in the previous results section. Since the least-squares approach has a well-established theoretical justification, the manuscript should do more to explore the advantages of Equation 6. This must be a pretty common issue in modelling. For example, how do ecosystem models of net primary productivity deal with this issue? I think the authors could do more to justify this new approach.

**AR: I'm not sure we agree with the reviewer on this point. In the context of calculating emission potentials from eddy covariance flux measurements the least squares approach has been used but to the best of our knowledge its use has never been explicitly justified. Indeed, the lack of justification was the partial**

motivation for this work. We would stress that the specific approach you take to calculate your "average" emission potential should depend on your proposed use of the BVOC model. In our manuscript we address this problem from the perspective of accounting, with the aim of producing an emission potential that allows us to properly simulate average or total emissions from a given forest over a given time period. We present eddy covariance flux measurements which we carefully correct for the effects of chemical flux divergence and dry deposition and therefore assume to represent the "best estimate" average emission from the site. Having established this "best estimate" average emission rate we can now use this to first back out an emission potential and secondly to challenge the model (combined with new emission potential). In practice this is no different from using a branch enclosure to measure the isoprene flux and then scaling to standard conditions using the algorithm to estimate the emission potential. Using this approach we have systematically evaluated various techniques for deriving a single "average" emission potential from a time series of flux measurements including through the LSR approach. As the reviewer suggests, the least squares approach has a well-established theoretical justification but only if a number of assumptions are fulfilled, two of which are that the data show a linear relationship and that the residuals are normally distributed. Figure 4 shows a plot of the measured isoprene flux versus the G93 γ term for the Bosco Fontana measurement site. It is clear that (i) the relationship is non-linear, driven by the algorithms inability to account for the attenuation of light and temperature through the canopy particularly during the periods either side of midday and (ii) the residuals are not normally-distributed. These two factors mean that the application of the LSR approach would be inappropriate. Indeed, our analysis shows that when the LSR method is used to estimate the "average" emission potential, then the algorithm subsequently fails to replicate the average observed flux. In contrast, adopting the "weighted average" approach ensures an emission potential with zero bias.

We will include this figure with short explanation in a revised version of the Supplementary Information and will emphasise our justification for choosing the weighted average method in the revised manuscript.

[Figure]

**Figure 4. Plot showing (a) the non-linearity in fluxes modelled using the G93 emission algorithm when compared with observations and (b) the distribution of residuals from a least square regression fit.**

Major comments

*Figures 2 and 3 pack in too much information. For example, I was interested in comparing the performance of the LSR & ODR approaches with MEGAN. In most cases in Figure 3, I could not distinguish these two cases because of overlapping plotting characters. What's the benefit of plotting all the different time average periods? Couldn't that be conveyed in a separate graph? Near lines 28-31 on page 12, you take away from Figure 3 that the G93 approach difference significantly from the MEGAN approach. This is well known, and could be conveyed more succinctly in a separate figure.

**AR: We will replot figure 3 so the points at each site are staggered horizontally to ensure all symbols are visible to the reader. In figure 2, where lines are masked by others we will change the covering lines to dashes.**

*The conclusion that "the emission potential is not constant throughout the day" should be refined. Within the modeling framework, the emission potential should be a constant throughout the day. The better way to frame this is that the calculated emission potential is not properly capturing the diurnal cycle. Also, considering just 08:00 to 18:00, there's not much variation in the EIP.

**AR: You are correct in the case of Figure 2, but the MEGAN algorithm didn't always perform so well. For example, Figures S1 and S4 showed much greater variation at the Alice Holt and O3HP sites respectively.**

*On lines 9-12, page 8, you mention the issue of the intercept for the least-squares approach. For the least-squares calculations in this paper, did you use a zero intercept?

**AR: No, in each case we did not force the intercept through zero as we felt this gives the regression only one degree of freedom. However, looking into this further we found that in most cases setting the intercept to zero only resulted in a very minor change to the calculated emission potential.**

Minor comments

*The abstract is a bit long. While comprehensive, I counted 660 words. In particular, some of the recommendations at the end repeat material from the abstract (factor of four). A target of 600 words seems more reasonable. With an open-access journal, there is less pressure on fitting so much in the abstract.

**AR: Agreed, we shorten the text, primarily through the removal of the last two sentences (which reviewer 2 did not approve of) and look to refine the text.**

Page 2, lines 33-34: The article by Arneth would be useful to consider and site at this point in the discussion (http://www.atmos-chem-phys.net/8/4605/2008/).

**AR: reference will be added**

Page 2, lines 34-36: Very minor point: branch enclosure measurements typically can't be performed at standard conditions. Instead, leaf temperature and light are measured, and often the Guenther algorithms are applied to derive a basal rate.

**AR: We will change this to just refer to leaf-level measurements**

Page 3, lines 5-7: Again, a good place to refer to Arneth et al 2008.

**AR: Reference will be added**

Page 3, line 22: Inconsistencies isn't the right notion here. Yes, there are inconsistencies, but there are also different assumptions.

**AR: This will be changed to "…inconsistencies and differences in the underlying assumptions…"**

Page 4, line 1: Since the algorithms for previous light and temperature are coming to come into play, some mention of the meteorological conditions during the campaigns compared to average climatology is necessary. In particular, where any of the campaigns conducted during times of water stress?

**AR: Agreed, where available we will add further details about the meteorological conditions at each site.**

Page 7, lines 21-32: This is a lot of text to describe something that wasn't used. Please consider if its necessary to include.

**AR: Although the PCEEA method was not shown in Figures 2 and 3, we do use it to derive emission potentials and the results are shown in Figures 4-6 and in the tables of emission potentials listed in the Supplementary Information. We therefore believe the brief description of the algorithm is merited.**

Page 8, line 25: Shouldn't this produce the same result as a linear regression with the intercept set to 0?

**AR: No, this is not the case because the datasets are never perfectly linear.**

Page 11, lines 16-18: "discernable" is subjective. This might be a real effect, or it might be random noise. Also, connect this to the major comment above: this variation represents a failure in the underlying model. Lines 26-27 (page 11) are the proper way to frame this conclusion.

**AR: We will remove the term "discernible"**

Page 12, lines 36-41: Yes, but this is only true when considering the extreme ends of the day. Typically, the focus is 10:00 – 16:00, when the variability is much lower with MEGAN.

**AR: You are correct in the case of Figure 2, but the MEGAN algorithm didn't always perform so well. For example, Figures S1 and S4 showed much greater variation at the Alice Holt and O3HP sites respectively, even within your suggested window of 10:00-16:00.**

Technical comments

Page 1, line 34: hyphenate 'site specific'

**AR: Done**

Page 2, line 18: hyphenate 'ground level'

**AR: Done**

Page 3, line 39: note explicitly this is ug of isoprene, not carbon (ugC), which has also been used in the past.

**AR: This will be changed to "…µg of isoprene m$^{-2}$ h$^{-1}$…"**

Page 4, line 13: be consistent about lat/long significant figures. The two used elsewhere are probably sufficient.

**AR: Changed**

Page 4, line 23: According to BG style, "32m platform".

**AR: Changed**

Page 7, line 7: hyphenate "in canopy"

**AR: Done**

Page 10, lines 6-7: fix grammar

**AR: Done**

Page 10, line 13: reflect should be reflects

**AR: Changed**

[revised manuscript text omitted]

**Supplementary Information**

**S1.1 Alice Holt – Measurement setup**

Above canopy-isoprene flux measurements at the Alice Holt forest site were made by combining fast measurements of isoprene made using a proton transfer reaction mass spectrometer (PTR-MS, Ionicon Analytik GmbH, Innsbruck, Austria), with measurements of the vertical wind velocity, made using a Gill Solent (R1012A) ultrasonic anemometer mounted atop a 25 m tall lattice tower at a height of 28.5 m. The PTR-MS was housed in a small container at the base of the tower and subsampled air from a 30 m PTFE tube (1/2" OD, 3/8" ID) which drew air from directly below the anemometer at a rate of 60 L min$^{-1}$ to ensure turbulent flow was achieved.

The PTR-MS operating conditions were held constant throughout the measurement period to ensure the reduced electric field strength ($E/N$, where $E$ is the electric field strength and $N$ is the buffer gas density) was maintained at 127 Td. The drift tube pressure, temperature and voltage were set to 2.01 mbar, 45 °C and 550 V respectively. When operating in flux mode the PTR-MS sequentially measured eight mass to charge ratios including the isotope of the primary ion ($m/z$ 21) and first water cluster ($m/z$ 37) which were both sampled at a rate 20 ms and $m/z$ 33, 45, 47, 59, 61, 69 and 71 which were all sampled at 50 ms. These dwell times are much shorter than is typical when measuring VOC fluxes by PTR-MS resulting in a lower signal-to-noise ratio than might be typical. This is because this campaign represented the first deployment of our flux system and therefore the optimal settings had not yet been determined. Here we only focus on the measurements of $m/z$ 69 which we attribute entirely to isoprene. Typically the ion counts reported by the PTR-MS are converted to a meaningful concentration by first calculating the instrument sensitivity to a specific compound determined by sampling from a gas standard. During the Alice Holt campaign no gas standard was available. Consequently, the recorded ion counts of isoprene per second ($I(RH^+)$) were converted to a measurement of isoprene concentration in units of parts per billion as follows

$$[R] = \frac{1}{k\Delta t} \frac{I(RH^+)}{T(RH^+)} \left( \frac{I(H_3O^+)}{T(H_3O^+)} \right)^{-1}, \tag{1}$$

where $I(RH^+)$ and $I(H_3O^+)$ are the isoprene and primary ion counts, respectively, $k$ is the reaction rate constant taken from Zhao and Zhang (2004) which was modified to account for the typical fragmentation of isoprene to $m/z$ 41 under the reported operating conditions and $\Delta t$ is the reaction term which is dependent upon the length of the reaction chamber. $T(RH^+)$ and $T(H_3O^+)$ are the instrument specific transmission efficiencies for isoprene and the primary ions. The transmission efficiencies were determined experimentally at the end of the measurement campaign. According to Taipale et al., (2008) the use of transmission efficiencies rather than instrument sensitivities calculated using gas standards can result in uncertainties of ~25%. The instrument background was measured once per day by sampling ambient air through a Pt/Al$_2$O$_3$ catalyst heated to 200 ºC and these values were subtracted from the ambient concentration measurements. Fluxes of isoprene were calculated following the procedures outlined by Langford et al. (2009). A cross-correlation between the vertical wind velocity and isoprene concentration was calculated for each averaging period to determine the time-lag between the two datasets which arises due to the spatial separation between ultrasonic anemometer and PTR-MS. Due to the low signal-to-noise ratio of the raw isoprene data (<10), the Following the recommendations of Langford et al. (2015) were followed to avoid systematic bias in the reported fluxes which involved the use of we calculated a prescribed time-lag which changed each day to reflect the average day-time (11:00 to 14:00) time-lag of that day.

**S1.2 Ispra – Measurement setup**

Isoprene flux measurements were made from June 11 to August 12, 2013 at the Ispra firest field station. The forest is further characterized with a different focus in Ferrea et al. (2012). More technical information on the general setup of the Ispra forest station can be found in Putaud et al. (2014).

For the turbulent flux measurements of isoprene, 10 Hz measurement data from a sonic anemometer (Gill, HS-100) were combined with 10 Hz concentration data from a fast isoprene sensor (FIS, Hills Scientific) mounted aloft a 37 m measurement tower . For the latter, air was drawn into a sampling line located 30 cm away from the sonic anemometer and carried at a flow rate of 25 slpm through a Teflon tube with 6 mm inner diameter to the FIS located inside an air conditioned container on the ground.

The FIS measurements are based on the detection of chemiluminescence occurring during the reaction of isoprene with ozone. Ambient air with a flow rate of 4-5 slpm and a 4 % mixture of ozone at 0.8 slpm in O$_2$ from an ozoniser (Hills Scientific) are mixed inside the reaction cell of the instrument. Following the reaction of isoprene with ozone, light is emitted at a characteristic wavelength and detected using single-photon counting at near-zero background. Instrument calibration to obtain

isoprene concentrations was done using zero air from a gas cylinder and air with certified isoprene concentrations on a weekly basis confirming practically no drift of the zero signal and little variation in the span during the measurement campaign.

The covariances between the high frequency wind data and isoprene concentration data were calculated using the EdiRe software package (University of Edinburgh). The median time lag between vertical wind speed and concentration measurements was 4.7 s with little fluctuation during the measurement campaign. This value was used in the final data processing.

**S2.    Isoprene Emission Potentials**

Ecosystem ($E_{eco}$), oak canopy ($E_{can}$) and leaf-level ($E_{LL}$) equivalent isoprene emission potentials (IEPs) and uncertainties for each of the five measurement sites are listed below. The IEPs were calculated using the six different implementations of the Guenther algorithm described in the manuscript. In each case the final IEP was determined using the weighted average IEP method.

**S2.1    Alice Holt**

Emission factors derived for Alice Holt are summarised in Tables S1 to S3.

**Table S1 Ecosystem-Scale isoprene emission potentials at Alice Holt**

| Algorithm | $E_{eco}$ | $E_{eco+Fd}$ | $E_{eco+Fd+chem}$ |
|---|---|---|---|
| G93 | 5613 | 6045 | 6347±1552 |
| G06 | 6542 | 7046 | 7398±1802 |
| PCEEA | 8368 | 9013 | 9464±2296 |
| MEGAN 2.1 (a) | 9333 | 10052 | 10555±2557 |
| MEGAN 2.1 (b) | 9686 | 10433 | 10955±2653 |
| MEGAN 2.1 (c) | 8781 | 9458 | 9931±2424 |

**Table S2 Oak canopy isoprene emission potentials at Alice Holt**

| Algorithm | $E_{can}$ | $E_{can+Fd}$ | $E_{can+Fd+chem}$ |
|---|---|---|---|
| G93 | 6237 | 6717 | 7053±2154 |
| G06 | 7269 | 7829 | 8220±2505 |
| PCEEA | 9298 | 10014 | 10515±3196 |
| MEGAN 2.1 (a) | 10370 | 11169 | 11727±3562 |
| MEGAN 2.1 (b) | 10762 | 11592 | 12172±3695 |
| MEGAN 2.1 (c) | 9757 | 10509 | 11034±3352 |

**Table S3 Leaf-level equivalent isoprene emission potentials at Alice Holt**

| Algorithm | $E_{LL}$ | $E_{LL+Fd}$ | $E_{LL+Fd+chem}$ |
|---|---|---|---|
| G93 | 74 | 80 | 84±31 |
| G06 | 77 | 83 | 87±32 |
| PCEEA | 98 | 106 | 111±41 |
| MEGAN 2.1 (a) | 110 | 118 | 124±46 |
| MEGAN 2.1 (b) | 114 | 123 | 129±47 |
| MEGAN 2.1 (c) | 103 | 111 | 117±43 |

**S2.1    Bosco Fontana**

Emission factors derived for Bosco Fontana are summarised in Tables S4 to S6.

**Table S4 Ecosystem-Scale isoprene emission potentials at Bosco Fontana**

| Algorithm | $E_{eco}$ | $E_{eco+Fd}$ | $E_{eco+Fd+chem}$ |
|---|---|---|---|
| G93 | 1529 | 1722 | 1791±440 |
| G06 | 720 | 810 | 843±375 |
| PCEEA | 1488 | 1675 | 1742±441 |
| MEGAN 2.1 (a) | 1376 | 1550 | 1612±428 |
| MEGAN 2.1 (b) | 1338 | 1507 | 1578±424 |
| MEGAN 2.1 (c) | 1980 | 2230 | 2319±493 |

**Table S5 Oak canopy isoprene emission potentials at Bosco Fontana**

| Algorithm | $E_{can}$ | $E_{can+Fd}$ | $E_{can+Fd+chem}$ |
|---|---|---|---|
| G93 | 5663 | 6378 | 6633±4002 |
| G06 | 2667 | 3000 | 3120±2212 |
| PCEEA | 5511 | 6204 | 6452±3906 |
| MEGAN 2.1 (a) | 5096 | 5741 | 5970±3648 |
| MEGAN 2.1 (b) | 4956 | 5581 | 5805±3560 |
| MEGAN 2.1 (c) | 7333 | 8259 | 8590±5069 |

**Table S6 Leaf-level equivalent isoprene emission potentials at Bosco Fontana**

| Algorithm | $E_{LL}$ | $E_{LL+Fd}$ | $E_{LL+Fd+chem}$ |
|---|---|---|---|
| G93 | 66 | 74 | 77±49 |
| G06 | 28 | 31 | 32±25 |
| PCEEA | 58 | 65 | 68±46 |
| MEGAN 2.1 (a) | 54 | 61 | 63±43 |
| MEGAN 2.1 (b) | 52 | 59 | 61±42 |
| MEGAN 2.1 (c) | 77 | 87 | 91±60 |

**S2.3 Castelporziano**

Emission factors derived for Bosco Fontana are summarised in Tables S7 to S9.

**Table S7 Ecosystem-Scale isoprene emission potentials at Castelporziano**

| Algorithm | $E_{eco}$ | $E_{eco+Fd}$ | $E_{eco+Fd+chem}$ |
|---|---|---|---|
| G93 | 99 | 106 | 111±14 |
| G06 | 66 | 70 | 74±911 |
| PCEEA | 122 | 130 | 137±16 |
| MEGAN 2.1 (a) | 107 | 115 | 121±15 |
| MEGAN 2.1 (b) | 110 | 117 | 123±14 |
| MEGAN 2.1 (c) | 144 | 155 | 163±19 |

10  **Table S8 Oak canopy isoprene emission potentials at Castelporziano**

| Algorithm | $E_{can}$ | $E_{can+Fd}$ | $E_{can+Fd+chem}$ |
|---|---|---|---|
| G93 | 360 | 385 | 405±232 |
| G06 | 240 | 255 | 267±156 |
| PCEEA | 444 | 473 | 496±285 |
| MEGAN 2.1 (a) | 389 | 418 | 439±251 |
| MEGAN 2.1 (b) | 400 | 425 | 447±257 |
| MEGAN 2.1 (c) | 524 | 564 | 592±337 |

**Table S9 Leaf-level equivalent isoprene emission potentials at Castelporziano**

| Algorithm | $E_{LL}$ | $E_{LL+Fd}$ | $E_{LL+Fd+chem}$ |
|---|---|---|---|
| G93 | 2.1 | 2.3 | 2.4±1.4 |
| G06 | 1.2 | 1.3 | 1.4±0.9 |
| PCEEA | 2.3 | 2.4 | 2.5±1.7 |
| MEGAN 2.1 (a) | 2.0 | 2.1 | 2.2±1.5 |
| MEGAN 2.1 (b) | 2.0 | 2.2 | 2.3±1.5 |

| | | | |
|---|---|---|---|
| MEGAN 2.1 (c) | 2.7 | 2.9 | 3.0±2.0 |

**S2.4 Ispra**

Emission factors derived for Ispra are summarised in Tables S10 to S12.

**Table S10 Ecosystem-Scale isoprene emission potentials at Ispra**

| Algorithm | $E_{eco}$ | $E_{eco+Fd}$ | $E_{eco+Fd+chem}$ |
|---|---|---|---|
| G93 | 5824 | 6385 | 6704±983 |
| G06 | 3591 | 3937 | 4133±748 |
| PCEEA | 6975 | 7646 | 8029±1120 |
| MEGAN 2.1 (a) | 6599 | 7234 | 7596±1074 |
| MEGAN 2.1 (b) | 6670 | 7312 | 7678±1082 |
| MEGAN 2.1 (c) | 8598 | 9426 | 9897±1321 |

**Table S11 Oak canopy isoprene emission potentials at Ispra**

| Algorithm | $E_{can}$ | $E_{can+Fd}$ | $E_{can+Fd+chem}$ |
|---|---|---|---|
| G93 | 7281 | 7981 | 8380±2073 |
| G06 | 4489 | 4921 | 5167±1391 |
| PCEEA | 8719 | 9558 | 10036±2443 |
| MEGAN 2.1 (a) | 8249 | 9042 | 9495±2321 |
| MEGAN 2.1 (b) | 8338 | 9140 | 9597±2344 |
| MEGAN 2.1 (c) | 10748 | 11782 | 12371±2969 |

**Table S12 Leaf-level equivalent isoprene emission potentials at Ispra**

| Algorithm | $E_{LL}$ | $E_{LL+Fd}$ | $E_{LL+Fd+chem}$ |
|---|---|---|---|
| G93 | 74 | 81 | 85±27 |
| G06 | 40 | 44 | 46±16 |
| PCEEA | 76 | 84 | 88±28 |
| MEGAN 2.1 (a) | 72 | 79 | 83±27 |
| MEGAN 2.1 (b) | 73 | 80 | 84±27 |
| MEGAN 2.1 (c) | 94 | 103 | 108±35 |

**S2.4 O3HP**

Emission factors derived for O3HP are summarised in Tables S13 to S15.

**Table S13 Ecosystem-Scale isoprene emission potentials at O3HP**

| Algorithm | $E_{eco}$ | $E_{eco+Fd}$ | $E_{eco+Fd+chem}$ |
|---|---|---|---|
| G93 | 5135 | 5642 | 5924±771 |
| G06 | 3439 | 3779 | 3967±551 |
| PCEEA | 7018 | 7710 | 8096±1026 |
| MEGAN 2.1 (a) | 6926 | 7610 | 7990±1014 |
| MEGAN 2.1 (b) | 7606 | 8357 | 8775±1107 |
| MEGAN 2.1 (c) | 8684 | 9541 | 10018±1255 |

**Table S14 Oak canopy isoprene emission potentials at O3HP**

| Algorithm | $E_{can}$ | $E_{can+Fd}$ | $E_{can+Fd+chem}$ |
|---|---|---|---|
| G93 | 6847 | 7523 | 7899±1945 |
| G06 | 4586 | 5038 | 5290±1328 |
| PCEEA | 9357 | 10280 | 10794±2639 |
| MEGAN 2.1 (a) | 9235 | 10146 | 10654±2605 |
| MEGAN 2.1 (b) | 10142 | 11142 | 11699±2857 |
| MEGAN 2.1 (c) | 11579 | 12721 | 13357±3256 |

**Table S15 Leaf-level equivalent isoprene emission potentials at O3HP**

| Algorithm | $E_{LL}$ | $E_{LL+Fd}$ | $E_{LL+Fd+chem}$ |
|---|---|---|---|
| G93 | 68 | 74 | 78±25 |
| G06 | 40 | 44 | 47±15 |
| PCEEA | 58 | 64 | 67±24 |
| MEGAN 2.1 (a) | 57 | 63 | 66±24 |
| MEGAN 2.1 (b) | 63 | 69 | 73±26 |
| MEGAN 2.1 (c) | 72 | 79 | 83±29 |

5 **S3    Comparison of isoprene emission potentials**

Tables S16 to S25 show a comparison of IEPs calculated at each of the five measurement sites using seven different methods to derive the average isoprene emission potential. All emission potentials shown have been corrected for deposition and chemical losses. The data in these tables forms the basis of Fig. 3 in the main manuscript.

**S3.1    Alice Holt, UK**

**Table S16 Comparison of isoprene emission potentials calculated using the MEGAN 2.1 (a) emission algorithm for Alice Holt in conjunction with the least square regression, orthogonal distance regression, weighted average and several variations of the midday**
15 **average methods.**

| | Fluxes | $\overline{IEP}$ (weighted) | $\overline{IEP}$ (all hours) | $\overline{IEP}$ (08 to 18) | $\overline{IEP}$ (10 to 15) | $\overline{IEP}$ (11 to 13) | LSR | ODR |
|---|---|---|---|---|---|---|---|---|
| IEP [µg m$^{-2}$ h$^{-1}$] | - | 10555 | 13251 | 11712 | 12316 | 12671 | 9349 | 12217 |
| Mean [µg m$^{-2}$ h$^{-1}$] | 779 | 779 | 978 | 864 | 909 | 935 | 690 | 902 |
| σ [µg m$^{-2}$ h$^{-1}$] | 1066 | **1097** | 1378 | 1218 | 1281 | 1317 | 972 | 1270 |
| r$^2$ | - | 0.54 | 0.54 | 0.54 | 0.54 | 0.54 | 0.54 | 0.54 |
| M score | - | 1.37 | 1.34 | 1.31 | 1.31 | 1.32 | 1.53 | **1.31** |
| Relative Bias [%] | - | **0** | 26 | 11 | 17 | 20 | -11 | 16 |

**Table S17 Comparison of isoprene emission potentials calculated using the G93 emission algorithm for Alice Holt in conjunction with the least square regression, orthogonal distance regression, weighted average and several variations of the midday average**
20 **methods.**

| | Fluxes | $\overline{IEP}$ (weighted) | $\overline{IEP}$ (all hours) | $\overline{IEP}$ (08 to 18) | $\overline{IEP}$ (10 to 15) | $\overline{IEP}$ (11 to 13) | LSR | ODR |
|---|---|---|---|---|---|---|---|---|
| IEP [µg m$^{-2}$ h$^{-1}$] | - | 6348 | 6062 | 6261 | 7607 | 8344 | 6995 | 7538 |
| Mean [µg m$^{-2}$ h$^{-1}$] | 779 | 779 | 744 | 768 | 933 | 1024 | 858 | 925 |
| σ [µg m$^{-2}$ h$^{-1}$] | 1327 | 915 | 874 | 902 | 1096 | **1203** | 1008 | 1086 |
| r$^2$ | 0.58 | 0.58 | 0.58 | 0.58 | 0.58 | 0.58 | 0.58 | 0.58 |
| M score | - | 1.239 | 1.315 | 1.260 | 1.065 | **1.054** | 1.121 | 1.069 |
| Relative Bias [%] | - | **0** | -4 | -1 | 20 | 31 | 10 | 19 |

**S3.2    Bosco Fontana, Italy**

**Table S18 Comparison of isoprene emission potentials calculated using the MEGAN 2.1 (a) emission algorithm for Bosco Fontana in conjunction with the least square regression, orthogonal distance regression, weighted average and several variations of the midday average methods.**

| | Fluxes | $\overline{IEP}$ (weighted) | $\overline{IEP}$ (all hours) | $\overline{IEP}$ (08 to 18) | $\overline{IEP}$ (10 to 15) | $\overline{IEP}$ (11 to 13) | LSR | ODR |
|---|---|---|---|---|---|---|---|---|
| IEP [µg m$^{-2}$ h$^{-1}$] | - | 1550 | 1493 | 1647 | 1509 | 1527 | 1489 | 1547 |
| Mean [µg m$^{-2}$ h$^{-1}$] | 862 | 862 | 830 | 916 | 839 | 849 | 828 | 860 |
| σ [µg m$^{-2}$ h$^{-1}$] | 1113 | 1053 | 1015 | 1119 | 1026 | 1038 | **1012** | 1052 |
| r$^2$ | - | 0.79 | 0.79 | 0.79 | 0.79 | 0.79 | 0.79 | 0.79 |
| M score | - | 0.347 | 0.356 | 0.347 | 0.353 | **0.350** | 0.357 | 0.347 |
| Relative Bias [%] | - | **0** | -4 | 6 | -3 | -1 | -4 | 0 |

**Table S19 Comparison of isoprene emission potentials calculated using the G93 emission algorithm for Bosco Fontana in conjunction with the least square regression, orthogonal distance regression, weighted average and several variations of the midday average methods.**

| | Fluxes | $\overline{IEP}$ (weighted) | $\overline{IEP}$ (all hours) | $\overline{IEP}$ (08 to 18) | $\overline{IEP}$ (10 to 15) | $\overline{IEP}$ (11 to 13) | LSR | ODR |
|---|---|---|---|---|---|---|---|---|
| IEP [µg m$^{-2}$ h$^{-1}$] | - | 1722 | 1229 | 1643 | 1996 | 2240 | 1953 | 1495 |
| Mean [µg m$^{-2}$ h$^{-1}$] | 862 | 862 | 615 | 822 | 999 | 1121 | 977 | 748 |
| σ [µg m$^{-2}$ h$^{-1}$] | 1113 | 854 | 609 | 815 | 990 | **1111** | 968 | 741 |
| r$^2$ | - | 0.75 | 0.75 | 0.75 | 0.75 | 0.75 | 0.75 | 0.75 |
| M score | - | 0.66 | 1.17 | 0.70 | **0.59** | 0.61 | 0.60 | 0.81 |
| Relative Bias [%] | - | **0** | -29 | -5 | 16 | 30 | 13 | -13 |

**S3.3 Castelporziano, Italy**

**Table S20 Comparison of isoprene emission potentials calculated using the MEGAN 2.1 (a) emission algorithm for Castelporziano in conjunction with the least square regression, orthogonal distance regression, weighted average and several variations of the midday average methods.**

| | Fluxes | $\overline{IEP}$ (weighted) | $\overline{IEP}$ (all hours) | $\overline{IEP}$ (08 to 18) | $\overline{IEP}$ (10 to 15) | $\overline{IEP}$ (11 to 13) | LSR | ODR |
|---|---|---|---|---|---|---|---|---|
| IEP [µg m$^{-2}$ h$^{-1}$] | - | 4374 | 4980 | 4680 | 3972 | 3874 | 3971 | 4782 |
| Mean [µg m$^{-2}$ h$^{-1}$] | 44 | 44 | 5048 | 4748 | 4044 | 3944 | 4043 | 4949 |
| σ [µg m$^{-2}$ h$^{-1}$] | 70 | 6157 | 6961 | 6561 | 5556 | 5456 | 5554 | 6763 |
| r$^2$ | - | 0.5554 | 0.5554 | 0.5554 | 0.5554 | 0.5554 | 0.5554 | 0.5554 |
| M score | - | 1.265 | 1.19 | 1.201.19 | 1.381.28 | 1.421.28 | 1.381.31 | 1.201.18 |
| Relative Bias [%] | - | **0** | 148 | 78 | -102 | -120 | -10-4 | 1011 |

**Table S21 Comparison of isoprene emission potentials calculated using the G93 emission algorithm for Castelporziano in conjunction with the least square regression, orthogonal distance regression, weighted average and several variations of the midday average methods.**

| | Fluxes | $\overline{IEP}$ (weighted) | $\overline{IEP}$ (all hours) | $\overline{IEP}$ (08 to 18) | $\overline{IEP}$ (10 to 15) | $\overline{IEP}$ (11 to 13) | LSR | ODR |
|---|---|---|---|---|---|---|---|---|

| | Fluxes | IEP (weighted) | IEP (all hours) | IEP (08 to 18) | IEP (10 to 15) | IEP (11 to 13) | LSR | ODR |
|---|---|---|---|---|---|---|---|---|
| **IEP** [µg m⁻² h⁻¹] | - | 103112 | 67105 | 11392 | 118114 | 13327 | 1193 | 11704 |
| **Mean** [µg m⁻² h⁻¹] | 44 | 44 | 4229 | 4540 | 4749 | 536 | 489 | 4745 |
| **σ** [µg m⁻² h⁻¹] | 70 | 4548 | 4529 | 4941 | 5150 | 577 | 510 | 5046 |
| **r²** | - | 0.4749 | 0.497 | 0.490.47 | 0.4947 | 0.497 | 0.497 | 0.497 |
| **M score** | - | 1.4841 | 1.512.72 | 1.3969 | 1.332 | **1.224** | 1.303 | 1.3445 |
| **Relative Bias [%]** | - | **0** | -534 | 2-10 | 612 | 1924 | 711 | 52 |

**S3.4  Ispra, Italy**

**Table S22** Comparison of isoprene emission potentials calculated using the MEGAN 2.1 (a) emission algorithm for Ispra in conjunction with the least square regression, orthogonal distance regression, weighted average and several variations of the midday average methods.

| | Fluxes | IEP (weighted) | IEP (all hours) | IEP (08 to 18) | IEP (10 to 15) | IEP (11 to 13) | LSR | ODR |
|---|---|---|---|---|---|---|---|---|
| **IEP** [µg m⁻² h⁻¹] | - | 7596 | 7212 | 7558 | 7928 | 8142 | 7504 | 9174 |
| **Mean** [µg m⁻² h⁻¹] | 2108 | 2108 | 2002 | 2098 | 2201 | 2261 | 2083 | 2546 |
| **σ** [µg m⁻² h⁻¹] | 3126 | 2940 | 2792 | 2925 | 3069 | 3152 | 2905 | 3551 |
| **r²** | - | 0.86 | 0.86 | 0.86 | 0.86 | 0.86 | 0.86 | 0.86 |
| **M score** | - | 0.28 | 0.32 | 0.29 | **0.27** | **0.27** | 0.29 | 0.31 |
| **Relative Bias [%]** | - | **0** | -5 | 0 | 4 | 7 | -1 | 21 |

**Table S23** Comparison of isoprene emission potentials calculated using the G93 emission algorithm for Ispra in conjunction with the least square regression, orthogonal distance regression, weighted average and several variations of the midday average methods.

| | Fluxes | IEP (weighted) | IEP (all hours) | IEP (08 to 18) | IEP (10 to 15) | IEP (11 to 13) | LSR | ODR |
|---|---|---|---|---|---|---|---|---|
| **IEP** [µg m⁻² h⁻¹] | - | 6703 | 5969 | 7629 | 7733 | 8359 | 7512 | 6966 |
| **Mean** [µg m⁻² h⁻¹] | 2108 | 2108 | 1877 | 2399 | 2432 | 2629 | 2363 | 2190 |
| **σ** [µg m⁻² h⁻¹] | 3126 | 2401 | 2139 | 2733 | 2771 | 2995 | 2691 | 2496 |
| **r²** | - | 0.78 | 0.78 | 0.78 | 0.78 | 0.78 | 0.78 | 0.78 |
| **M score** | - | 0.51 | 0.65 | 0.44 | **0.43** | 0.44 | 0.44 | 0.48 |
| **Relative Bias [%]** | - | **0** | -11 | 14 | 15 | 25 | 12 | 4 |

**S3.5  O3HP, France**

**Table S24** Comparison of isoprene emission potentials calculated using the MEGAN 2.1 (a) emission algorithm for O3HP in conjunction with the least square regression, orthogonal distance regression, weighted average and several variations of the midday average methods.

| | Fluxes | IEP (weighted) | IEP (all hours) | IEP (08 to 18) | IEP (10 to 15) | IEP (11 to 13) | LSR | ODR |
|---|---|---|---|---|---|---|---|---|
| **IEP** [µg m⁻² h⁻¹] | | 7991 | 6914 | 7795 | 7883 | 7889 | 8138 | 8018 |

| Mean [µg m$^{-2}$ h$^{-1}$] | 899 | 899 | 777 | 877 | 886 | 887 | 915 | 902 |
|---|---|---|---|---|---|---|---|---|
| σ [µg m$^{-2}$ h$^{-1}$] | 1371 | 1279 | 1107 | 1247 | 1262 | 1262 | **1302** | 1283 |
| r$^2$ | - | 0.90 | 0.90 | 0.90 | 0.90 | 0.90 | 0.90 | 0.90 |
| M score | - | 0.23 | 0.34 | 0.24 | 0.23 | 0.23 | **0.22** | 0.23 |
| Relative Bias [%] | - | **0** | 0 | -13 | -2 | -1 | 0 | 2 |

**Table S25 Comparison of isoprene emission potentials calculated using the G93 emission algorithm for O3HP in conjunction with the least square regression, orthogonal distance regression, weighted average and several variations of the midday average methods.**

| | Fluxes | IEP (weighted) | IEP (all hours) | IEP (08 to 18) | IEP (10 to 15) | IEP (11 to 13) | LSR | ODR |
|---|---|---|---|---|---|---|---|---|
| IEP [µg m$^{-2}$ h$^{-1}$] | | 5924 | 6894 | 5607 | 6576 | 6902 | 7225 | 5513 |
| Mean [µg m$^{-2}$ h$^{-1}$] | 899 | 899 | 1046 | 851 | 998 | 1047 | 1096 | 836 |
| σ [µg m$^{-2}$ h$^{-1}$] | 1371 | 1031 | 1200 | 977 | 1145 | 1201 | **1258** | 960 |
| r$^2$ | - | 0.84 | 0.84 | 0.84 | 0.84 | 0.84 | 0.84 | 0.84 |
| M score | - | 0.43 | 0.34 | 0.49 | 0.36 | 0.34 | **0.34** | 0.52 |
| Relative Bias [%] | - | **0** | 16 | -5 | 11 | 16 | 22 | -7 |

**S4    Emission potential calculation assessment**

Figures S1 to S4 show the average diurnal profile of the isoprene emission potential that have been calculated by inverting the
5  G93 (Panel A) and MEGAN 2.1 (a) (Panel C) emission algorithms. Also shown are the average emission potential assigned to
each site which were calculated using seven different methods (see main text for details).

**S4.1    Alice Holt**

[Figure]

[Figure]

**Figure S1 Panels A and C show the average diurnal cycle in the isoprene emission potential (e.g. $IEP = \overline{\left(\frac{F_{iso}}{\gamma}\right)}$) calculated for the Alice Holt site, UK using the G93 (panel A) and MEGAN 2.1 (panel B) algorithms. Superimposed on top of these are the isoprene emission potentials calculated using the least square regression, orthogonal distance regression and average (with several averaging lengths) methods – see text for detailed description. Panels B and D show the average diurnal cycle of the measured fluxes and the average diurnal cycle of the fluxes modelled using the seven different isoprene emission potentials calculated for this data set.**

**S4.2    Bosco Fontana**

[Figure]

[Figure]

**Figure S2 Panels A and C show the average diurnal cycle in the isoprene emission potential (e.g. $IEP = \overline{\left(\frac{F_{iso}}{\gamma}\right)}$)    calculated for the Bosco Fontana site, Italy using the G93 (panel A) and MEGAN 2.1 (panel B) algorithms. Superimposed on top of these are the isoprene emission potentials calculated using the least square regression, orthogonal distance regression and average (with several averaging lengths) methods – see text for detailed description. Panels B and D show the average diurnal cycle of the measured fluxes and the average diurnal cycle of the fluxes modelled using the seven different isoprene emission potentials calculated for this data set.**

**S4.3  Castelporziano**

[Figure]

[Figure]

**Figure S3 Panels A and C show the average diurnal cycle in the isoprene emission potential (e.g.** $IEP = \left(\overline{\frac{F_{iso}}{\gamma}}\right)$**) calculated for the Castelporziano site, Italy using the G93 (panel A) and MEGAN 2.1 (panel B) algorithms. Superimposed on top of these are the isoprene emission potentials calculated using the least square regression, orthogonal distance regression and average (with several averaging lengths) methods – see text for detailed description. Panels B and D show the average diurnal cycle of the measured fluxes and the average diurnal cycle of the fluxes modelled using the seven different isoprene emission potentials calculated for this data set.**

**S4.4    O3HP**

[Figure]

**Figure S4 Panels A and C show the average diurnal cycle in the isoprene emission potential (e.g. $IEP = \overline{\left(\frac{F_{iso}}{\gamma}\right)}$)    calculated for the Observatoire de Haute Provance site, France using the G93 (panel A) and MEGAN 2.1 (panel B) algorithms. Superimposed on top of these are the isoprene emission potentials calculated using the least square regression, orthogonal distance regression and average (with several averaging lengths) methods – see text for detailed description. Panels B and D show the average diurnal cycle of the**

**measured fluxes and the average diurnal cycle of the fluxes modelled using the seven different isoprene emission potentials calculated for this data set.**

**S5    Influence of past light and temperature on calculated emission potential**

the influence of past light and temperature on derived emission potentials is considered relative to a set of standard within canopy, leaf-level, conditions. When experimentalists use the MEGAN model in a big-leaf approach, these leaf-level conditions are typically calculated using measurements of PPFD and temperature made above the forest canopy. Figures S5 to S9 show the time series of the average 24 hour and 240 hour above canopy  PPFD and temperature for each of the five sites relative to the leaf-level standard conditions used in MEGAN. Typically the past 24 and 240 hour PPFD is considerably higher than the standard conditions which results in a much larger gamma factor and a reduced emission potential. For this reason, it is our recommendation that the MEGAN model should not be used to derive emission potentials unless coupled to an appropriate canopy environment model.

**S5.1    Alice Holt**

[Figure]

[Figure]

[Figure]

**Figure S5.** Time series of the previous (24 and 240 hours) above canopy light and temperature measurements made at the Alice Holt site relative to the standard conditions used in the Model of Emission of Gases and Aerosols from Nature (dashed lines) for leaf-level canopy average temperature and light.

**S5.2    Bosco Fontana**

[Figure]

[Figure]

**Figure S6.** Time series of the previous (24 and 240 hours) above canopy light and temperature measurements made at the Bosco Fontana site relative to the standard conditions used in the Model of Emission of Gases and Aerosols from Nature (dashed lines) for leaf-level canopy average temperature and light.

**S5.3    Castelporziano**

[Figure]

[Figure]

**Figure S7.** Time series of the previous (24 and 240 hours) above canopy light and temperature measurements made at the Castelporziano site relative to the standard conditions used in the Model of Emission of Gases and Aerosols from Nature (dashed lines) for leaf-level canopy average temperature and light.

**S5.4    Ispra**

[Figure]

[Figure]

**Figure S8.** Time series of the previous (24 and 240 hours) above canopy light and temperature measurements made at the Ispra forest site relative to the standard conditions used in the Model of Emission of Gases and Aerosols from Nature (dashed lines) for leaf-level canopy average temperature and light.

**S5.5    O3HP**

[Figure]

[Figure]

**Figure S9.** Time series of the previous (24 and 240 hours) above canopy light and temperature measurements made at the Observatoire de Haute Provence site relative to the standard conditions used in the Model of Emission of Gases and Aerosols from Nature (dashed lines) for leaf-level canopy average temperature and light.

**S6 Species composition uncertainty**

Species composition data for each of the five measurement sites was available, but no uncertainties were stated. Here, we attempt to assess the uncertainty by performing wind rose analysis of the isoprene emission potentials to assess how they vary spatially. Figure S10 shows the isoprene emission potential wind rose for the Alice Holt, Bosco Fontana and Ispra forest sites

10  relative to the average isoprene emission potential. This analysis showed the spatial variation to range from 14% at Alice Holt to 20% at Bosco Fontana. For the remaining two sites, Castelporziano and O3HP, where there was insufficient data to perform a detailed wind rose analysis, uncertainties of 20% were assigned.

[Figure]

[Figure]

[Figure]

Figure S10 Calculated isoprene emission potential by wind sector (red) and site average (blue) for the Alice Holt (a), Ispra (b) and Bosco Fontana (c) measurement sites.

**S7 Reporting fluxes for a set of defined conditions**

Above canopy flux measurements may be used to determine an emission potential for a specific set of defined conditions. We used two-dimensional histograms to determine the most common set of light and temperature conditions observed during day time at each of the five measurement sites. The histograms binned the number of flux averaging periods that corresponded to specific bin ranges of light ($\pm$ 200 µmol m$^{-2}$ s$^{-1}$) and temperature ($\pm$ 1 K). The results for the are shown in figures S11 to S14

[Figure]

Figure S11 Two dimensional histogram plot of flux averaging periods that correspond to bins of light ($\pm$ 200 $\mu$mol m$^{-2}$ s$^{-1}$) and temperature ($\pm$ 1 K) at the Alice Holt measurement site.

[Figure]

Figure S12 Two dimensional histogram plot of flux averaging periods that correspond to bins of light (± 200 µmol m⁻² s⁻¹) and temperature (± 1 K) at the Bosco Fontana measurement site.

[Figure]

[Figure]

Figure S13 Two dimensional histogram plot of flux averaging periods that correspond to bins of light ($\pm$ 200 µmol m$^{-2}$ s$^{-1}$) and temperature ($\pm$ 1 K) at the Castelporziano measurement site.

[Figure]

Figure S14 Two dimensional histogram plot of flux averaging periods that correspond to bins of light (± 200 µmol m$^{-2}$ s$^{-1}$) and temperature (± 1 K) at the O3HP measurement site.